



# Chempath 1.0: An open-source pathway analysis program for photochemical models

Daniel Garduno Ruiz[1], Colin Goldblatt[1], and Anne-Sofie Ahm[1]

[1]School of Earth and Ocean Sciences, University of Victoria, Victoria, British Columbia, Canada

**Correspondence:** Daniel Garduno Ruiz (danielgardunoruiz@uvic.ca)

**Abstract.** We describe the development of *Chempath*: an open-source pathway analysis program for photochemical models. This algorithm can help understand the results of complex photochemical models by identifying the most important reaction chains (pathways) for the production and destruction of a species of interest in a reaction system. The algorithm can also quantify the contribution of the pathways to the production and destruction of a species.

We demonstrate how to apply *Chempath* to a one-dimensional photochemical model, using an example of a reaction system for Earth's present-day atmosphere. We validate that *Chempath* can identify well-known chemical mechanisms for $O_3$ production and destruction in this model, suggesting that this algorithm can be applied to understand photochemical models of less well-known atmospheres, like past and exoplanet atmospheres.

## 1 Introduction

The construction of chemical pathways is essential to understand the results of photochemical models. These models typically represent hundreds of reactions producing and destroying chemical species within the atmosphere (see for example Hu et al. 2012; Tsai et al. 2017; Wogan et al. 2022). The interaction of these reactions makes it difficult to attribute the production or destruction of a species to a single reaction. Instead, to understand the mechanisms that affect the concentration of a species it is necessary to construct pathways. A pathway is a sequence of reactions that interact with each other to produce, destroy, or

recycle a species. For example, stratospheric ozone destruction is explained in terms of pathways that catalyze $O_3$ destruction (Lary, 1997). One of these pathways involves the reaction of $O_3$ with chlorine species (Molina and Rowland, 1974):

$$O_3 + hv \longrightarrow O_2 + O$$
$$Cl + O_3 \longrightarrow ClO + O_2$$
$$ClO + O \longrightarrow Cl + O_2$$
$$Net: 2\,O_3 \longrightarrow 3\,O_2$$

(ClO pathway)

This ozone-destroying pathway has three reactions, and its net effect is to convert two $O_3$ molecules into three $O_2$ molecules.

  Several methods can help understand the results of photochemical models. For example, sensitivity analyses can constrain

uncertainties in reaction rate constants and provide information on how variable the results of a model are when the reaction





rates are perturbed (Turányi and Tomlin, 2014). Wiring diagrams can help understand the flow of molecules in a reaction system (Fishtik et al., 2006; Androulakis, 2006). However, these methods can not give quantitative information about the chains of reactions responsible for the production or destruction of a species in a chemical model. To understand these chemical mechanisms, it is necessary to construct pathways.

Chemical pathways are usually constructed manually and empirically, tracking and connecting reactions important for the production or destruction of a species of interest. This approach is the most widely used for pathway construction. However, the manual construction of pathways can not give a quantitative estimate of how important a pathway is for the production or destruction of a species relative to other pathways. Also, the manual construction of pathways has reproducibility limitations. Alternatively, pathways can be automatically constructed using algorithms (Milner, 1964; Schuster and Schuster, 1993; Clarke, 30    1988; Lehmann, 2002, 2004).

One of the most used algorithms to construct pathways is the "Pathway analysis program" created by Lehmann (2004). This algorithm can automatically construct all the significant pathways in a reaction system and calculate the contribution of each pathway to the production and destruction of a species of interest. The "Pathway analysis program" has been used in several studies to gain a better understanding of atmospheric chemistry models (Grenfell et al., 2006; Verronen et al., 2011; Stock 35    et al., 2012a, b; Verronen and Lehmann, 2013; Stock et al., 2017; Gebauer et al., 2017). However, none of these studies provide instructions to reproduce their results, and they do not offer an open-source code to apply the algorithm to other problems. The lack of an open-source pathway analysis program limits the application of this algorithm to photochemical models.

In this paper, we describe the development of *Chempath*: an open-source implementation of Lehmann's (2004) algorithm for analysis of photochemical models. We aim to contribute this open-source pathway analysis program to enhance the applicability 40    of this algorithm to photochemical models and to enhance the replicability of chemical pathway construction. We demonstrate how to apply this algorithm to the one-dimensional photochemical model *photochem* (Wogan et al. 2023, https://github.com/ Nicholaswogan/photochem). *Photochem* is an updated version of *Photochempy* (Wogan, 2023), a model that has been used for exoplanet and early Earth photochemical studies (Wogan et al., 2022; Thompson et al., 2022; Garduno Ruiz et al., 2023, 2024). This model originates from *Atmos*, a photochemical model extensively used to investigate photochemistry in exoplanet and 45    past atmospheres (see for example Kasting et al. 1979; Kasting and Donahue 1980; Segura et al. 2005; Zahnle et al. 2006; Claire et al. 2014; Arney et al. 2016).

The structure of the paper is as follows. In section 2, we review the Lehmann (2004) algorithm via a simple example. In section 3, we describe how we implemented *Chempath*. In section 4, we demonstrate how to apply *Chempath* to the one-dimensional photochemical model *photochem*.

## 50  2  Algorithm review

Here we provide a summary of the pathway analysis program presented in Lehmann (2004), using a simple example to explain each step of the algorithm (see the original paper for further details). The pathway analysis program forms pathways by the iterative connection of reactions through branching-point species (figure 1). A branching-point species is one that is used to





connect reactions that produce the species with reactions that destroy it. For example, the reaction $Cl + O_3 \longrightarrow ClO + O_2$ can
be connected with the reaction $ClO + O \longrightarrow Cl + O_2$ through the branching-point species $ClO$.

The example we use to demonstrate the algorithm consists of five reactions between five species involving hydrogen oxide
radicals. The reactions are:

$(R_1)$ $H_2O_2 + OH \longrightarrow HO_2 + H_2O$, with rate=1 ppb/hr,

$(R_2)$ $OH + OH \longrightarrow H_2O_2$, with rate=0.5 ppb/hr,

$(R_3)$ $H_2O_2 + O \longrightarrow OH + HO_2$, with rate=1.5 ppb/hr,

$(R_4)$ $H_2O + hv \longrightarrow OH + O$, with rate=5 ppb/hr,

$(R_5)$ $H_2O_2 + hv \longrightarrow OH + OH$, with rate=0.1 ppb/hr,

and the species are:

$(S_1)$ $H_2O_2$, with a concentration of 3 ppb,

$(S_2)$ $OH$, with a concentration of 6 ppb,

$(S_3)$ $HO_2$, with a concentration of 2 ppb,

$(S_4)$ $H_2O$, with a concentration of 10000 ppb,

$(S_5)$ $O$, with a concentration of 10 ppb,

We arbitrarily select these rates and concentrations for this example. Assuming this reaction system was run for 1 hour, the
production and destruction by the reactions result in concentration changes of $-2.1$, $4.7$, $2.5$, $-4$, and $3.5$ ppb for each species
respectively. In the next sections, we are going to identify which combination of reactions is the most important to explain the
$HO_2$ concentration change.

## 2.1 Assumptions and definitions

The algorithm uses the variables listed in table 1 to construct pathways. In all the variables $i = 1, \ldots, n_s, j = 1, \ldots, n_r$ and
$k = 1, \ldots, n_p$ where $n_s$ is the number of species, $n_r$ is the number of reactions and $n_p$ is the number of pathways.

| Variable | Initial value in simple example | Units in simple example | Description |
| --- | --- | --- | --- |





| | | | Units | Description |
|---|---|---|---|---|
| $s_{ij}$ | $\begin{array}{ccccc} R_1 & R_2 & R_3 & R_4 & R_5 \\ \end{array}$ $\begin{pmatrix} -1 & 1 & -1 & 0 & -1 \\ -1 & -2 & 1 & 1 & 2 \\ 1 & 0 & 1 & 0 & 0 \\ 1 & 0 & 0 & -1 & 0 \\ 0 & 0 & -1 & 1 & 0 \end{pmatrix}$ $\begin{array}{c} S_1 \\ S_2 \\ S_3 \\ S_4 \\ S_5 \end{array}$ | | ppb | Matrix representing the number of molecules of species $S_i$ produced ($s_{ij} > 0$) or destroyed ($s_{ij} < 0$) by reaction $R_j$. For example, in the simple example reaction $R_5$ destroys one molecule of species $S_1$, so $s_{15} = -1$. |
| $dt$ | 1 | | hr | Time step. |
| $dc_i$ | $\begin{array}{ccccc} S_1 & S_2 & S_3 & S_4 & S_5 \\ \end{array}$ $\begin{bmatrix} -2.1 & 4.7 & 2.5 & -4 & 3.5 \end{bmatrix}$ | | ppb | Change in concentration of species $S_i$ in the time step $dt$. |
| $\bar{c}_i$ | $\begin{array}{ccccc} S_1 & S_2 & S_3 & S_4 & S_5 \\ \end{array}$ $\begin{bmatrix} 1.95 & 8.35 & 3.25 & 9998 & 11.75 \end{bmatrix}$ | | ppb | Mean concentration of species $S_i$ in the time step $dt$. |
| $\delta_i = \frac{dc_i}{dt}$ | $\begin{array}{ccccc} S_1 & S_2 & S_3 & S_4 & S_5 \\ \end{array}$ $\begin{bmatrix} -2.1 & 4.7 & 2.5 & -4 & 3.5 \end{bmatrix}$ | | ppb/hr | Rate of concentration change of species $S_i$. |
| $f_k$ | $\begin{array}{ccccc} P_1 & P_2 & P_3 & P_4 & P_5 \\ \end{array}$ $\begin{bmatrix} 1 & 0.5 & 1.5 & 5 & 0.1 \end{bmatrix}$ | | ppb/hr | Rate of pathway $P_k$. |
| $r_j$ | $\begin{array}{ccccc} R_1 & R_2 & R_3 & R_4 & R_5 \\ \end{array}$ $\begin{bmatrix} 1 & 0.5 & 1.5 & 5 & 0.1 \end{bmatrix}$ | | ppb/hr | Mean rate of reaction $R_j$ in time step $dt$. |
| $\tilde{r}_j$ | $\begin{array}{ccccc} R_1 & R_2 & R_3 & R_4 & R_5 \\ \end{array}$ $\begin{bmatrix} 0 & 0 & 0 & 0 & 0 \end{bmatrix}$ | | ppb/hr | Part of the rate of reaction $R_j$ associated with deleted pathways. |
| $\tilde{p}_i$ | $\begin{array}{ccccc} S_1 & S_2 & S_3 & S_4 & S_5 \\ \end{array}$ $\begin{bmatrix} 0 & 0 & 0 & 0 & 0 \end{bmatrix}$ | | ppb/hr | Rate of production of species $S_i$ by deleted pathways. |
| $\tilde{d}_i$ | $\begin{array}{ccccc} S_1 & S_2 & S_3 & S_4 & S_5 \\ \end{array}$ $\begin{bmatrix} 0 & 0 & 0 & 0 & 0 \end{bmatrix}$ | | ppb/hr | Rate of destruction of species $S_i$ by deleted pathways. |





| Formula | Matrix | Units | Description |
|---|---|---|---|
| $p_i = \tilde{p}_i + \sum_{k=1}^{n_p} m_{ik} \cdot f_k$ for $m_{ik} > 0$ | $\begin{array}{ccccc} S_1 & S_2 & S_3 & S_4 & S_5 \end{array}$ $\begin{bmatrix} 0.5 & 6.7 & 2.5 & 1 & 5 \end{bmatrix}$ | ppb/hr | Rate of production of species $S_i$ by all pathways (including deleted pathways). |
| $d_i = \tilde{d}_i + \sum_{k=1}^{n_p} \lvert m_{ik} \rvert \cdot f_k$ for $m_{ik} < 0$ | $\begin{array}{ccccc} S_1 & S_2 & S_3 & S_4 & S_5 \end{array}$ $\begin{bmatrix} 2.6 & 2 & 0 & 5 & 1.5 \end{bmatrix}$ | ppb/hr | Rate of destruction of species $S_i$ by all pathways (including deleted pathways). |
| $x_{jk}$ | $\begin{array}{ccccc} P_1 & P_2 & P_3 & P_4 & P_5 \end{array}$ $\begin{pmatrix} 1 & 0 & 0 & 0 & 0 \\ 0 & 1 & 0 & 0 & 0 \\ 0 & 0 & 1 & 0 & 0 \\ 0 & 0 & 0 & 1 & 0 \\ 0 & 0 & 0 & 0 & 1 \end{pmatrix} \begin{array}{c} R_1 \\ R_2 \\ R_3 \\ R_4 \\ R_5 \end{array}$ | No units | Matrix representing the multiplicity of reaction $R_j$ in pathway $P_k$. A multiplicity is the number of times a reaction occurs in a pathway. For example, in the ClO pathway presented above all reactions have multiplicities equal to 1. If a reaction does not occur in a pathway, $x_{jk} = 0$. Initially, this matrix is set equal to an identity matrix with the size of the number of reactions $n_r$. |
| $m_{ik} = \sum_{j=1}^{n_r} s_{ij} \cdot x_{jk}$ | $\begin{array}{ccccc} P_1 & P_2 & P_3 & P_4 & P_5 \end{array}$ $\begin{pmatrix} -1 & 1 & -1 & 0 & -1 \\ -1 & -2 & 1 & 1 & 2 \\ 1 & 0 & 1 & 0 & 0 \\ 1 & 0 & 0 & -1 & 0 \\ 0 & 0 & -1 & 1 & 0 \end{pmatrix} \begin{array}{c} S_1 \\ S_2 \\ S_3 \\ S_4 \\ S_5 \end{array}$ | ppb | Matrix representing the number of molecules of species $S_i$ produced ($m_{ik} > 0$) or destroyed ($m_{ik} < 0$) by pathway $P_k$. This variable is equal to the matrix multiplication of $s_{ij}$ and $x_{jk}$. |





| $\tau_i = \frac{\bar{c_i}}{d_i}$ | $\begin{matrix} S_1 & S_2 & S_3 & S_4 & S_5 \end{matrix}$ $\begin{bmatrix} 0.75 & 4.175 & \text{Nan} & 1999.6 & 7.83 \end{bmatrix}$ | hr | Lifetime of species $S_i$. If the destruction by all pathways $d_i$ is zero, the lifetime becomes undefined (Nan). This means that there are no pathways consuming $S_i$. |
|---|---|---|---|

Table 1: Variables used in the pathway analysis algorithm and their initial values in the simple example used to explain the algorithm. In all the variables $i = 1, \ldots, n_s, j = 1, \ldots, n_r$ and $k = 1, \ldots, n_p$ where $n_s$ is the number of species, $n_r$ is the number of reactions and $n_p$ is the number of pathways.

The algorithm assumes that the reactions are unidirectional and that the user splits the reversible reactions into their forward and backward components. It is also assumed that mass is conserved in the analyzed chemical model, and the reactions produce the exact number of molecules to explain the concentration changes:

$$dc_i = \sum_{j=1}^{n_r} (s_{ij} \cdot r_j)dt, \quad i = 1 \ldots n_s. \tag{1}$$

For example, in our simple example we can verify that this condition is satisfied for $HO_2$ ($S_3$):

$$dc_3 = \sum_{j=1}^{5} (s_{3j} \cdot r_j)dt = \sum_{j=1}^{5} ([1,0,1,0,0] \cdot [1,0.5,1.5,5,0.1]\text{ppb/hr}) \, 1\text{hr} = 2.5\text{ppb} \tag{2}$$

### 2.2 Algorithm initialization

The algorithm requires four inputs from a chemical kinetics model in two consecutive time steps: the reaction system with $n_r$ reactions between $n_s$ species, the species concentrations, the mean reaction rates, and the model time in these two consecutive

time steps. The user can also input a minimum rate of pathways $f_{\min}$. With these inputs, the algorithm determines all pathways with a rate $> f_{\min}$. In the simple example, we are going to use a minimum rate of pathways $f_{\min} = 0.02\text{ppb/hr}$.

The algorithm uses the input information to initialize the variables listed in table 1. At first, each pathway is considered to have only one reaction, and the matrix $x_{jk}$ is initialized as an identity matrix with the size of the number of reactions. This means that initially, pathway $P_1$ only contains reaction $R_1$, pathway $P_2$ only contains reaction $R_2$, etc. The rates of the

pathways are initialized with the rates of the reactions: $f_{k\,\text{init}} = r_j$. The variables $\tilde{r}_j$, $\tilde{p}_i$, and $\tilde{d}_i$ that store rates associated with deleted pathways are initialized as arrays of zeros.

### 2.3 Choice on branching-point species

Once the algorithm has been initialized, the next step is to choose a branching point species $S_b$ to start constructing pathways. Species with lower lifetimes are selected as branching-point species first because a small lifetime is associated with fast



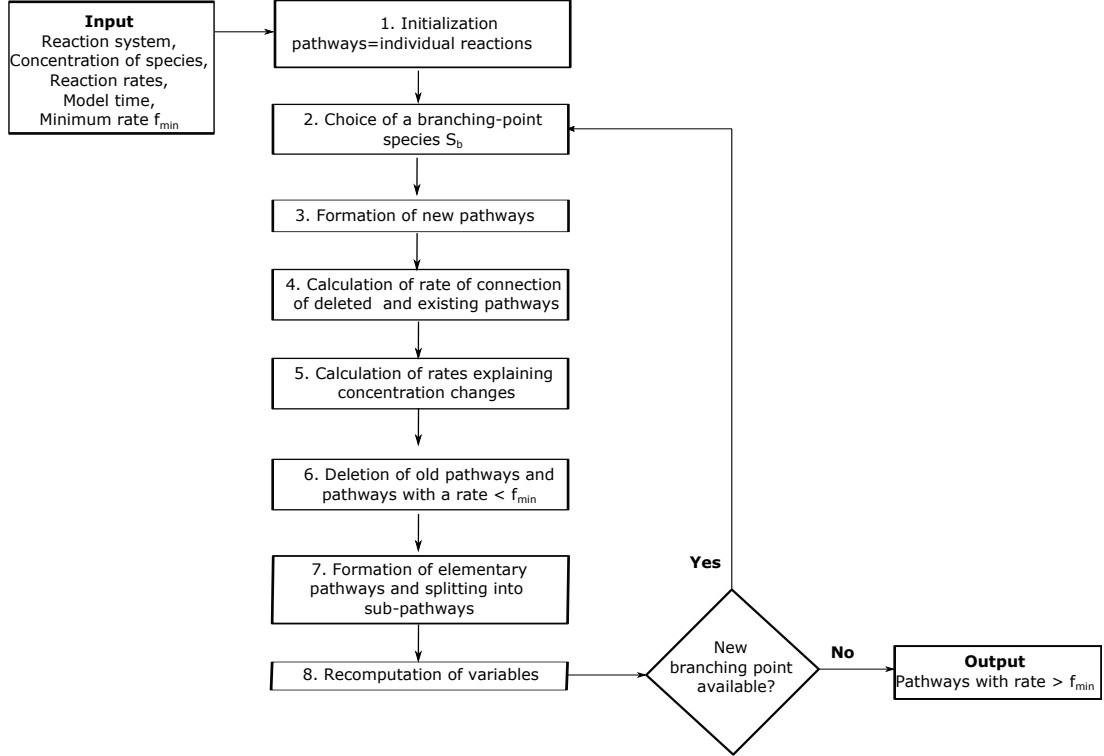

**Figure 1.** Pathway analysis Program algorithm. For a full description of the algorithm see Lehmann (2004).

consumption by reactions. The algorithm forms new pathways through the iterative connection of previously formed pathways through branching-point species until there are no more branching-point species left (figure 1). A species can be a branching-point species only once. To find pathways that destroy or produce a specific species of interest, the species itself is not used as a branching-point. Otherwise, the pathways producing and consuming this species would be connected. Sometimes it is useful to treat some species as inert or long-lived, not considering them as branching-point species. For example, in the atmosphere $N_2$ is long-lived and inert, so it could not be considered as a branching-point.

In the simple example, where we are investigating the change in $HO_2$, we are going to treat $H_2O$ as a long-lived inert species, not considering it as a branching-point. The species with the lowest lifetime is $H_2O_2$ ($\tau_1 = 0.75hr$), so this species is the first branching-point.

## 2.4 Formation of new pathways and rate calculations

The algorithm forms new pathways connecting all previously formed pathways that produce a branching-point species $S_b$ with all pathways that destroy $S_b$. The connections are made ensuring that the new pathways do not produce nor destroy the branching-point species $S_b$. If a pathway $P_k$ produces $m_{bk}$ molecules of the branching-point species $S_b$ and a pathway $P_l$ destroys $m_{bl}$ molecules of $S_b$, the connection of these pathways forms a new pathway $P_n$ with multiplicities:





$$x_{jn} = |m_{bl}|x_{jk} + m_{bk}x_{jl} \tag{3}$$

where $j = 1 \ldots n_r$, $k$ and $l$ are the indexes of the pathways producing and destroying $S_b$, and $b$ is the index of the species $S_b$. The rate $f_n$ of the new pathway $P_n$ is calculated by multiplying the rates of the producing ($f_k$) and destructing ($f_l$) pathways and dividing by the maximum of the rate of production and destruction of the branching-point species by all pathways:

$$f_n = \frac{f_k f_l}{\max(p_b, d_b)}, \tag{4}$$

where $k$ and $l$ are the indexes of the pathways producing and destroying the branching-point species $S_b$, and $b$ is the index
of the branching-point species. This equation is derived by calculating the rate at which the molecules of $S_b$ formed by $P_k$ are destroyed by $P_l$ (see Lehmann 2004 for the derivation). One can think of equation 4 as distributing the rate of a pathway to new pathways using the probability that a molecule consumed or produced by one pathway is consumed or produced by another pathway. For example, if the change in concentration of the branching-point species $dc_b > 0$, then the chemical production is greater than the chemical destruction ($p_b > d_b$), and equation 4 takes the form $f_n = \frac{f_k f_l}{p_b}$. The ratio $\pi = \frac{f_k}{p_b}$ can be interpreted
as the probability that a molecule of the branching-point species is produced by the pathway $P_k$. Since the pathway $P_l$ is going to be connected with all pathways $P_k$ producing $S_b$, and the sum of the probabilities $\pi$ over all pathways producing the branching-point species $S_b$ is equal to one, the rate $f_l$ is going to be completely distributed to the new pathways.

The multiplicities $x_{jn}$ of the new pathways are divided by their greatest common divisor $g$ to keep them as simple as possible. The rates of the new pathways are multiplied by $g$ to avoid altering the number of molecules that the new pathways produce or
destroy. The new pathways and their rates are appended to $x_{jk}$ and $f_k$ respectively.

In the simple example, there are three pathways destroying the branching-point species $H_2O_2$ ($P_1$, $P_3$ and $P_5$), and one pathway producing it ($P_2$). The connection of these pathways will result in three new pathways. For example, the pathway $P_1$ destroys one $H_2O_2$ molecule ($m_{11} = -1$), and the pathway $P_2$ produces one $H_2O_2$ molecule ($m_{12} = 1$), so the connection of these pathways results in a new pathway with multiplicities:

$$x_{jn} = |m_{11}|x_{j2} + m_{12}x_{j1} = [0,1,0,0,0] + [1,0,0,0,0] = [1,1,0,0,0]. \tag{5}$$

The rate of this new pathway is:

$$f_n = \frac{f_2 f_1}{\max(p_1, d_1)} = \frac{0.5\text{ppb/hr} \cdot 1\text{ppb/hr}}{2.6\text{ppb/hr}} = 0.192\text{ppb/hr} \tag{6}$$

After forming all new pathways at the branching-point species $H_2O_2$, the multiplicities $x_{jk}$ and the pathway rates $f_k$ will have a form similar to:





$$
x_{jk} = \begin{array}{c} \\ \begin{array}{cccccccc} P_1 & P_2 & P_3 & P_4 & P_5 & P_6 & P_7 & P_8 \end{array} \\ \begin{pmatrix} 1 & 0 & 0 & 0 & 0 & 1 & 0 & 0 \\ 0 & 1 & 0 & 0 & 0 & 1 & 1 & 1 \\ 0 & 0 & 1 & 0 & 0 & 0 & 1 & 0 \\ 0 & 0 & 0 & 1 & 0 & 0 & 0 & 0 \\ 0 & 0 & 0 & 0 & 1 & 0 & 0 & 1 \end{pmatrix} \begin{array}{c} R_1 \\ R_2 \\ R_3 \\ R_4 \\ R_5 \end{array} \end{array}
\tag{7}
$$

$$
f_k = \begin{array}{c} \begin{array}{cccccccc} P_1 & P_2 & P_3 & P_4 & P_5 & P_6 & P_7 & P_8 \end{array} \\ \begin{bmatrix} 1 & 0.5 & 1.5 & 5 & 0.1 & 0.192 & 0.288 & 0.0192 \end{bmatrix} \end{array} \text{ppb/hr}
$$

We can think of a column in $x_{jk}$ as a pathway. For example, the pathway that we formed before ($P_6$) is located in the 6th column of the matrix $x_{jk}$ in equation 7, and contains one times reaction $R_1$ and one times reaction $R_2$:

$$
\begin{aligned}
\mathrm{H_2O_2 + OH} &\longrightarrow \mathrm{HO_2 + H_2O} \\
\mathrm{OH + OH} &\longrightarrow \mathrm{H_2O_2} \\
\text{Net:} \, \mathrm{3\,OH} &\longrightarrow \mathrm{HO_2 + H_2O}
\end{aligned}
\tag{$P_6$}
$$

### 2.5 Calculation of rate of connection of deleted and existing pathways

In this step, the algorithm calculates the rates of the connection of deleted pathways with existing pathways. Pathways with a rate lower than $f_{\min}$ will be deleted in a subsequent step (see section 2.7). In the first iteration of the algorithm there are no deleted pathways, but in other iterations some pathways could have been deleted by this point. In that case, the deleted pathways can not be connected with existing pathways in the previous step. However, the algorithm keeps track of the rates of production and destruction of the branching-point species $S_b$ that would have been computed in these connections. The deleted pathways produce the branching-point species $S_b$ at a rate $\tilde{p}_b$, so according to equation 4, the rate of the connection of deleted pathways that produce $S_b$ with existing pathways $P_e$ that destroy $S_b$ is:

$$
\tilde{f}_e = \frac{f_e \tilde{p}_b}{\max(p_b, d_b)},
\tag{8}
$$

where $e$ represents the indexes of the pathways that destroy the branching-point species $S_b$. Similarly, deleted pathways consuming $S_b$ at a rate $\tilde{d}_b$ would have been connected with existing pathways $P_e$ producing $S_b$ at a rate:

$$
\tilde{f}_e = \frac{f_e \tilde{d}_b}{\max(p_b, d_b)},
\tag{9}
$$





where $e$ represents the indexes of the pathways that produce $S_b$. These rates are added to the the variables $\tilde{r}_j$, $\tilde{p}_i$ and $\tilde{d}_i$ that store rates associated with deleted pathways:

$$\tilde{r}_j = \tilde{r}_j + \sum_k x_{je} \cdot \tilde{f}_e$$

$$\tilde{p}_i = \tilde{p}_i + \sum_k m_{ie} \tilde{f}_e \quad \text{for } m_{ie} > 0$$

$$\tilde{d}_i = \tilde{d}_i + \sum_k |m_{ie}| \tilde{f}_e \quad \text{for } m_{ie} < 0$$

(10)

where $i = 1 \ldots n_s, j = 1 \ldots n_r, e$ = indexes of pathways producing or destroying the branching-point species $S_b$. In the sim-
ple example, there are no deleted pathways yet, so $\tilde{f}_e = 0$.

### 2.6 Calculation of rates explaining concentration changes

In this step, the algorithm redefines the rates of the pathways that contribute to the concentration change of the branching-point species $S_b$. The pathways that produce molecules of the branching point species $S_b$ contribute to its concentration change $dc_b$ if $dc_b > 0$. Similarly, the pathways that destroy molecules of the branching point species $S_b$ contribute to its concentration
change $dc_b$ if $dc_b < 0$. The algorithm redefines the rate $f_k$ of these pathways, keeping the fraction $\hat{f}_k$ of $f_k$ that contributes to the concentration change $dc_b$. This fraction is calculated by multiplying the rate $f_k$ by the absolute value of the rate of concentration change of the branching point species $\delta_b$, and dividing by the maximum of the production and destruction of the branching-point species by all pathways:

$$\hat{f}_k = \frac{f_k |\delta_b|}{\max(p_b, d_b)}$$

(11)

where $k$ = indexes of pathways producing or destroying $S_b$, and $b$ is the index of the branching-point species $S_b$. This rate is derived by calculating the probability that a molecule of $S_b$ produced or destroyed by a pathway contributes to the concentration change of the branching point species $dc_b$ (see Lehmann 2004 for the derivation). If $dc_b = 0$ there is no redefinition of rates.

In the simple example, the branching-point species $H_2O_2$ has a concentration change $dc_1 = -2.1$ppb. Since $dc_1 < 0$, we are going to redefine the rates of the pathways that destroy $H_2O_2$ because they contribute to the concentration change $dc_1$. For
example, the pathway $P_1$ has a single reaction destroying $H_2O_2$ at a rate of 1ppb/hr. The part of this rate that contributes to the $H_2O_2$ concentration change is:

$$\hat{f}_1 = \frac{f_1 |\delta_1|}{\max(p_1, d_1)} = \frac{1\text{ppb/hr} \cdot 2.1\text{ppb/hr}}{2.6\text{ppb/hr}} = 0.807\text{ppb/hr}.$$

(12)

After all the rates of the pathways destroying $S_b$ are redefined, $f_k$ has the form:





$$f_k = \begin{array}{cccccccc} P_1 & P_2 & P_3 & P_4 & P_5 & P_6 & P_7 & P_8 \\ \left[0.807\right. & 0.5 & 1.211 & 5 & 0.0807 & 0.192 & 0.288 & \left.0.0192\right] \end{array} \text{ ppb/hr} \tag{13}$$

## 2.7 Deletion of old pathways and pathways with a rate < $f_{\min}$

After a pathway producing the branching-point species $S_b$ has been connected with all pathways destroying $S_b$, it is eliminated from the matrix $x_{jk}$ if it does not contribute to the concentration change $dc_b$. If $dc_b > 0$, the pathways that destroy the branching-point species are deleted. If $dc_b < 0$, the pathways that produce the branching-point species are deleted, and if $dc_b = 0$, both the pathways that produce and destroy the branching-point species are deleted. In this case, the variables $\tilde{r}_j$, $\tilde{p}_i$, and $\tilde{d}_i$ that store the effect of deleted pathways are not updated because the rates of these pathways were completely distributed to new pathways.

In the simple example $dc_1 < 0$, so the pathways $P_1$, $P_3$ and $P_5$ that were used to form new pathways will not be deleted because they destroy molecules of the branching-point species $H_2O_2$, contributing to its concentration change. However, the pathway $P_2$ does not contribute to $dc_1$, so it is deleted.

The algorithm also deletes pathways with a rate $< f_{\min}$ to avoid constructing an unmanageable number of pathways and to enhance the computing time. If there are $n_q$ pathways with rates $f_q < f_{\min}$, these pathways are deleted from the matrix $x_{jk}$ and the the variables $\tilde{r}_j$, $\tilde{p}_i$ and $\tilde{d}_i$ are updated according to:

$$\tilde{r}_j = \tilde{r}_j + \sum_q x_{jq} \cdot f_q$$
$$\tilde{p}_i = \tilde{p}_i + \sum_q m_{iq} \cdot f_q \quad \text{for } m_{iq} > 0 \tag{14}$$
$$\tilde{d}_i = \tilde{d}_i + \sum_q |m_{iq}| \cdot f_q \quad \text{for } m_{iq} < 0$$

where $i = 1 \ldots n_s, j = 1 \ldots n_r$, and $q =$ indexes of pathways with rate $< f_{\min}$. The rates of these pathways are also deleted from the array $f_k$.

In the simple example, the pathway $P_8$ (column 8 of $x_{jk}$ in equation 7) has a rate lower than the minimum rate of pathways $f_{\min} = 0.02$ppb/hr, so this pathway will be deleted and the variables that store the information of deleted pathways will be updated using equations 14. For example, $\tilde{r}_j$ will be updated according to:

$$\tilde{r}_j = \tilde{r}_j + x_{j8} \cdot f_8 = [0,0,0,0,0] + [0,1,0,1,0] \cdot 0.0192\text{ppb/hr} = [0,0.0192,0,0.0192,0]\text{ppb/hr}. \tag{15}$$

Since pathway $P_8$ is a null cycle with reactions $R_2$ and $R_5$, the number of molecules of species $S_i$ produced or destroyed by this pathway $m_{i8} = 0$, and $\tilde{p}_i$ and $\tilde{d}_i$ will not be affected by equations 14. After these deletions, the multiplicities $x_{jk}$ and the pathway rates $f_k$ will have a form similar to:





$$
x_{jk} =
\begin{array}{cccccc}
P_1 & P_2 & P_3 & P_4 & P_5 & P_6
\end{array}
\begin{pmatrix}
1 & 0 & 0 & 0 & 1 & 0 \\
0 & 0 & 0 & 0 & 1 & 1 \\
0 & 1 & 0 & 0 & 0 & 1 \\
0 & 0 & 1 & 0 & 0 & 0 \\
0 & 0 & 0 & 1 & 0 & 0
\end{pmatrix}
\begin{array}{l}
R_1 \\ R_2 \\ R_3 \\ R_4 \\ R_5
\end{array}
\tag{16}
$$

$$
f_k =
\begin{array}{cccccc}
P_1 & P_2 & P_3 & P_4 & P_5 & P_6
\end{array}
\begin{bmatrix}
0.807 & 1.211 & 5 & 0.0807 & 0.192 & 0.288
\end{bmatrix} \ \text{ppb/hr}
$$

Note that when pathways are deleted from the matrix $x_{jk}$ there is a redefinition of pathways. For example, since pathway
$P_2$ was deleted, now there is a new pathway $P_2$ that corresponds to the second column of $x_{jk}$ in equation 16.

### 2.8 Formation of elementary pathways and splitting into sub-pathways

The steps above can produce pathways with a large and unnecessary number of reactions. The algorithm splits these complex
pathways into shorter, simpler pathways. The first step in this process is to find the elementary sub-pathways of a complex
pathway. A pathway $P_s$ is a sub-pathway of a pathway $P_c$ if all the reactions in $P_s$ are in $P_c$. Elementary pathways do not have
sub-pathways. The algorithm finds the elementary sub-pathways of a pathway $P_c$ by forming new pathways with the reactions
contained in $P_c$ and keeping only the elementary pathways (see Lehmann 2004 for a full description of this process). If a
pathway $P_c$ with multiplicities $x_{jc}$ has $n_e$ elementary pathways with multiplicities $x'_{je}, e = 1 \ldots n_e$ , the algorithm splits $x_{jc}$
into the sub-pathways $x'_{je}$ finding weighs $w_e$ that fulfill the equation:

$$
x_{jc} = \sum_{e=1}^{n_e} w_e x'_{je}, \text{ where } j = 1 \ldots n_r \text{ and } c = \text{ index of pathway to be split.} \tag{17}
$$

The rate $f_c$ of the pathway $P_c$ is distributed to the sub-pathways using the weighs $w_e$:

$$
f_e = f_c w_e, \text{ where } e = 1 \ldots n_e, \text{ and } c = \text{ index of pathway to be split.} \tag{18}
$$

After finding the sub-pathways, the algorithm deletes $x_{jc}$ from $x_{jk}$ and appends the new-sub pathways $x'_{je}$ into $x_{jk}$. Sim-
ilarly, the rate $f_c$ is deleted, and the rates $f_e$ are appended to $f_k$. If a sub-pathway already exists in the matrix $x_{jk}$, its rate is
added to the already existing pathway.
As noted by Lehmann (2004), equation 17 can have multiple solutions, leading to slightly different results according to
the solution that one chooses. However, these differences tend to be small and the overall result of the algorithm is similar



even when equation 17 has multiple solutions (Lehmann, 2004). We solve equation 17 using Scipy's "lsq_linear" function, minimizing the equation:

$$0.5||Ax - b||^2 \text{ with constraints } 0 \leq x \leq \infty, \tag{19}$$

where $||x||$ is the norm of $x$, $A = x_{jc}$, $x = w_e$ and $b = x'_{je}$. Equation 19 is a convex optimization problem that is guaranteed to have a global minimum solution. Thus, when there are multiple solutions to equation 17, we choose the solution that minimizes the most equation 19.

In the first iteration of the algorithm in the simple example there are no pathways to split. See section 2.10 for an example of how to split pathways into sub-pathways.

## 2.9 Re-computation of variables

The variables $m_{ik}$, $p_i$, and $d_i$ are recomputed using the definitions presented in table 1 to match the information from the new pathways formed in the above steps. This re-computation is done after deleting pathways and after splitting pathways into sub-pathways.

In the simple example, this re-computation results in the following values for $m_{ik}$, $p_i$, and $d_i$:

$$m_{ik} = \sum_{j=1}^{5} \overbrace{\begin{pmatrix} -1 & 1 & -1 & 0 & -1 \\ -1 & -2 & 1 & 1 & 2 \\ 1 & 0 & 1 & 0 & 0 \\ 1 & 0 & 0 & -1 & 0 \\ 0 & 0 & -1 & 1 & 0 \end{pmatrix}}^{s_{ij}} \overbrace{\begin{pmatrix} 1 & 0 & 0 & 0 & 1 & 0 \\ 0 & 0 & 0 & 0 & 1 & 1 \\ 0 & 1 & 0 & 0 & 0 & 1 \\ 0 & 0 & 1 & 0 & 0 & 0 \\ 0 & 0 & 0 & 1 & 0 & 0 \end{pmatrix}}^{x_{jk}} = \begin{array}{c} \begin{array}{cccccc} P_1 & P_2 & P_3 & P_4 & P_5 & P_6 \end{array} \\ \begin{pmatrix} -1 & -1 & 0 & -1 & 0 & 0 \\ -1 & 1 & 1 & 2 & -3 & -1 \\ 1 & 1 & 0 & 0 & 1 & 1 \\ 1 & 0 & -1 & 0 & 1 & 0 \\ 0 & -1 & 1 & 0 & 0 & -1 \end{pmatrix} \begin{array}{c} S_1 \\ S_2 \\ S_3 \\ S_4 \\ S_5 \end{array} \end{array} \tag{20}$$

$$p_i = \overbrace{\begin{bmatrix} 0 & 0 & 0 & 0 & 0 \end{bmatrix}}^{\tilde{p}_i} + \sum_{k=1}^{6} \overbrace{\begin{pmatrix} 0 & 0 & 0 & 0 & 0 & 0 \\ 0 & 1 & 1 & 2 & 0 & 0 \\ 1 & 1 & 0 & 0 & 1 & 1 \\ 1 & 0 & 0 & 0 & 1 & 0 \\ 0 & 0 & 1 & 0 & 0 & 0 \end{pmatrix}}^{m_{ik} \text{ for } m_{ik} > 0} \overbrace{\begin{pmatrix} 0.807 \\ 1.211 \\ 5 \\ 0.0807 \\ 0.192 \\ 0.288 \end{pmatrix}}^{f_k} = \begin{array}{ccccc} S_1 & S_2 & S_3 & S_4 & S_5 \end{array} \\ \begin{bmatrix} 0 & 6.373 & 2.5 & 1 & 5 \end{bmatrix} \text{ ppb/hr} \tag{21}$$





$$d_i = \overset{\tilde{d}_i}{\begin{bmatrix} 0 & 0 & 0 & 0 & 0 \end{bmatrix}} + \sum_{k=1}^{6} \overset{|m_{ik}| \text{ for } m_{ik} < 0}{\begin{pmatrix} 1 & 1 & 0 & 1 & 0 & 0 \\ 1 & 0 & 0 & 0 & 3 & 1 \\ 0 & 0 & 0 & 0 & 0 & 0 \\ 0 & 0 & 1 & 0 & 0 & 0 \\ 0 & 1 & 0 & 0 & 0 & 1 \end{pmatrix}} \overset{f_k}{\begin{pmatrix} 0.807 \\ 1.211 \\ 5 \\ 0.0807 \\ 0.192 \\ 0.288 \end{pmatrix}} = \overset{S_1 \quad S_2 \quad S_3 \quad S_4 \quad S_5}{\begin{bmatrix} 2.1 & 1.673 & 0 & 5 & 1.5 \end{bmatrix}} \text{ ppb/hr} \quad (22)$$

### 2.10 Second iteration in simple example

After the first iteration at the branching point species $H_2O_2$ in the simple example, we ended up with 6 pathways (equation 16). The species with the smallest lifetime with respect to these new pathways and the next branching-point species is OH ($S_2$). Looking at the second row of $m_{ik}$ in equation 20 ($m_{2k}$), we can see that pathways $P_1, P_5$ and $P_6$ destroy OH and pathways $P_2$, $P_3$ and $P_4$ produce OH. The connection of these pathways will lead to the formation of 9 new pathways (section 2.4). For example, connecting pathways $P_2$ and $P_5$ we obtain the pathway:


$$x_{jn} = |m_{25}|x_{j2} + m_{22}x_{j5} = 3 \cdot [0,0,1,0,0] + 1 \cdot [1,1,0,0,0] = [1,1,3,0,0]$$

$$f_n = \frac{f_2 f_5}{\max(p_2, d_2)} = \frac{1.211\text{ppb/hr} \cdot 0.192\text{ppb/hr}}{6.373\text{ppb/hr}} = 0.036\text{ppb/hr}$$

(23)

  At this point there is no production or destruction of OH by deleted pathways, so the rates of connection of existing pathways with deleted pathways $\tilde{f}_e = 0$ (section 2.5). Since the concentration change of the branching-point species OH is $dc_2 = 4.7ppb > 0$, the rates of the pathways producing OH are redefined to keep the fraction that contributes to $dc_2$ (section 2.6), and the pathways that destroy OH are deleted because they do not contribute to $dc_2$ (section 2.7). Three pathways with

rate $< f_{\min}$ are also deleted (section 2.7).

  The next step is to split pathways into sub-pathways (section 2.8). The pathway formed above (equation 23) contains one times reaction $R_1$, one times reaction $R_2$ and three times reaction $R_3$:

$$\begin{aligned} H_2O_2 + OH &\longrightarrow HO_2 + H_2O \\ OH + OH &\longrightarrow H_2O_2 \\ 3\,(H_2O_2 + O &\longrightarrow OH + HO_2) \\ \hline \text{Net:}\ 3\,H_2O_2 + 3\,O &\longrightarrow 4\,HO_2 + H_2O \end{aligned} \qquad (P_n)$$

  This pathway can be split into two simpler pathways:





$$OH + OH \longrightarrow H_2O_2$$

$$2\,(H_2O_2 + O \longrightarrow OH + HO_2) \tag{$P_{e1}$}$$

$$Net\!: H_2O_2 + 2\,O \longrightarrow 2\,HO_2$$

$$H_2O_2 + OH \longrightarrow HO_2 + H_2O$$

$$H_2O_2 + O \longrightarrow OH + HO_2 \tag{$P_{e2}$}$$

$$Net\!: 2\,H_2O_2 + O \longrightarrow 2\,HO_2 + H_2O$$

In this case, the solution to equation 17 is $w_e = [1,1]$ and $P_n = P_{e1} + P_{e2}$. The sub-pathways have the same rate as the initial pathway because the weights $w_e$ are equal to 1. The sub-pathways $P_{e1}$ and $P_{e2}$ were formed before, when the connection of pathways that produce OH with pathways that destroy OH was made, so their rate is going to be added to the rate of the already

existing pathways, and the initial pathway will be deleted from $x_{jk}$.

At the end of this iteration, the variables $x_{jk}$, $f_k$, $m_{ik}$, $p_i$, $d_i$, $\tilde{p}_i$, $\tilde{d}_i$ and $\tilde{r}_j$ will have a form similar to:

$$x_{jk} = \begin{array}{c} \\ \\ \\ \\ \\ \end{array}\begin{pmatrix} P_1 & P_2 & P_3 & P_4 & P_5 & P_6 & P_7 & P_8 \\ 0 & 0 & 0 & 1 & 0 & 1 & 1 & 0 \\ 0 & 0 & 0 & 0 & 1 & 0 & 1 & 1 \\ 1 & 0 & 0 & 1 & 2 & 0 & 0 & 1 \\ 0 & 1 & 0 & 0 & 0 & 1 & 3 & 1 \\ 0 & 0 & 1 & 0 & 0 & 0 & 0 & 0 \end{pmatrix}\begin{array}{l} \\ R_1 \\ R_2 \\ R_3 \\ R_4 \\ R_5 \end{array} \tag{24}$$

$$f_k = \begin{array}{c} P_1 & P_2 & P_3 & P_4 & P_5 & P_6 & P_7 & P_8 \\ \left[\, 0.893 & 3.687 & 0.0595 & 0.19 & 0.091 & 0.633 & 0.15 & 0.226 \,\right] \end{array} \; \text{ppb/hr} \tag{25}$$


$$m_{ik} = \begin{array}{c} \\ \\ \\ \\ \\ \end{array}\begin{pmatrix} P_1 & P_2 & P_3 & P_4 & P_5 & P_6 & P_7 & P_8 \\ -1 & 0 & -1 & -2 & -1 & -1 & 0 & 0 \\ 1 & 1 & 2 & 0 & 0 & 0 & 0 & 0 \\ 1 & 0 & 0 & 2 & 2 & 1 & 1 & 1 \\ 0 & -1 & 0 & 1 & 0 & 0 & -2 & -1 \\ -1 & 1 & 0 & -1 & -2 & 1 & 3 & 0 \end{pmatrix}\begin{array}{l} \\ S_1 \\ S_2 \\ S_3 \\ S_4 \\ S_5 \end{array} \tag{26}$$





$$p_i = \begin{array}{ccccc} S_1 & S_2 & S_3 & S_4 & S_5 \\ \begin{bmatrix} 0 & 4.7 & 2.5 & 0.215 & 4.773 \end{bmatrix} \end{array} \quad \text{ppb/hr} \tag{27}$$

$$d_i = \begin{array}{ccccc} S_1 & S_2 & S_3 & S_4 & S_5 \\ \begin{bmatrix} 2.1 & 0.0 & 0.0 & 4.215 & 1.273 \end{bmatrix} \end{array} \quad \text{ppb/hr} \tag{28}$$

$$\tilde{p}_i = \begin{array}{ccccc} S_1 & S_2 & S_3 & S_4 & S_5 \\ \begin{bmatrix} 0 & 0 & 0.032 & 0.025 & 0 \end{bmatrix} \end{array} \quad \text{ppb/hr} \tag{29}$$

$$\tilde{d}_i = \begin{array}{ccccc} S_1 & S_2 & S_3 & S_4 & S_5 \\ \begin{bmatrix} 0.0417 & 0.0 & 0.0 & 0.0 & 0.0073 \end{bmatrix} \end{array} \quad \text{ppb/hr} \tag{30}$$

$$\tilde{r}_j = \begin{array}{ccccc} R_1 & R_2 & R_3 & R_4 & R_5 \\ \begin{bmatrix} 0.025 & 0.031 & 0.007 & 0.0 & 0.04 \end{bmatrix} \end{array} \quad \text{ppb/hr} \tag{31}$$

### 2.11 Final iteration in simple example

The final branching-point species in the simple example is O ($S_5$). In this final iteration of the algorithm there are eight pathways in the matrix $x_{jk}$ (equation 24). Looking at the 5th row of $m_{ik}$ ($m_{5k}$) we can see that the pathways $P_1$, $P_4$ and $P_5$ consume O and the pathways $P_2$, $P_6$ and $P_7$ produce O. The connection of these pathways will lead to the formation of nine new pathways.

At this point, the deleted pathways destroy 0.007 O ppb/hr (equation 30). This means that we will need to account for the connection of deleted pathways with existing pathways (section 2.5). For example, the pathway $P_2$ with rate $f_2 = 3.687\text{ppb/hr}$ produces one molecule of O. This pathway would have been connected with the deleted pathways at a rate (equation 10):

$$\tilde{f}_2 = \frac{f_2 \tilde{d}_5}{\max(p_5, d_5)} = \frac{3.687\text{ppb/hr} \cdot 0.007\text{ppb/hr}}{4.773\text{ppb/hr}} = 0.005\text{ppb/hr} \tag{32}$$

This rate is used to update the variables that store the deleted rates (equation 10):





$$\tilde{r}_j = \tilde{r}_j + x_{j2} \cdot \tilde{f}_2 = [0.025, 0.031, 0.007, 0.0, 0.04] + [0,0,0,1,0] \cdot 0.005 \text{ppb/hr}$$

$$= \begin{array}{ccccc} R_1 & R_2 & R_3 & R_4 & R_5 \\ \left[0.025 \quad 0.031 \quad 0.007 \quad 0.005 \quad 0.04\right] \end{array} \text{ ppb/hr}$$

$$\tilde{p}_i = \tilde{p}_i + m_{i2}\tilde{f}_2 \quad \text{for } m_{i2} > 0 = [0,0,0.032,0.025,0] + [0,1,0,0,1] \cdot 0.005 \text{ppb/hr}$$

$$= \begin{array}{ccccc} S_1 & S_2 & S_3 & S_4 & S_5 \\ \left[0 \quad 0.005 \quad 0.032 \quad 0.03 \quad 0.005\right] \end{array} \text{ ppb/hr} \tag{33}$$

$$\tilde{d}_i = \tilde{d}_i + |m_{i2}|\tilde{f}_2 \quad \text{for } m_{i2} < 0 = [0.0417, 0.0, 0.0, 0.0, 0.0073] + [0,0,0,1,0] \cdot 0.005 \text{ppb/hr}$$

$$= \begin{array}{ccccc} S_1 & S_2 & S_3 & S_4 & S_5 \\ \left[0.0417 \quad 0.0 \quad 0.0 \quad 0.005 \quad 0.0073\right] \end{array} \text{ ppb/hr}$$

The same operations are done for pathways $P_6$ and $P_7$. After accounting for the connection of deleted and existing pathways, the rates of the pathways $P_2$, $P_6$ and $P_7$ are redefined because they contribute to the $O$ concentration change $dc_5 = 3.5 ppb$

(section 2.6). The pathways $P_1$, $P_4$, and $P_5$ are deleted because they do not contribute to the $O$ concentration change, and three pathways with rate $< f_{\min}$ are deleted (section 2.7). Three pathways are split into sub-pathways (section 2.8), and at the end of this iteration, the variables $x_{jk}$, $f_k$, $m_{ik}$, $p_i$, $d_i$, $\tilde{p}_i$ and $\tilde{d}_i$ will have a form similar to:

$$x_{jk} = \begin{array}{ccccccc} P_1 & P_2 & P_3 & P_4 & P_5 & P_6 & P_7 \\ \begin{pmatrix} 0 & 0 & 1 & 1 & 0 & 0 & 2 \\ 0 & 0 & 0 & 1 & 1 & 0 & 0 \\ 0 & 0 & 0 & 0 & 1 & 1 & 1 \\ 1 & 0 & 1 & 3 & 1 & 1 & 1 \\ 0 & 1 & 0 & 0 & 0 & 0 & 0 \end{pmatrix} & \begin{array}{c} R_1 \\ R_2 \\ R_3 \\ R_4 \\ R_5 \end{array} \end{array} \tag{34}$$

$$f_k = \begin{array}{ccccccc} P_1 & P_2 & P_3 & P_4 & P_5 & P_6 & P_7 \\ \left[2.704 \quad 0.06 \quad 0.465 \quad 0.111 \quad 0.325 \quad 0.936 \quad 0.172\right] \end{array} \text{ ppb/hr} \tag{35}$$





$$
m_{ik} = \begin{array}{c} \begin{array}{ccccccc} P_1 & P_2 & P_3 & P_4 & P_5 & P_6 & P_7 \end{array} \\ \left( \begin{array}{ccccccc} 0 & -1 & -1 & 0 & 0 & -1 & -3 \\ 1 & 2 & 0 & 0 & 0 & 2 & 0 \\ 0 & 0 & 1 & 1 & 1 & 1 & 3 \\ -1 & 0 & 0 & -2 & -1 & -1 & 1 \\ 1 & 0 & 1 & 3 & 0 & 0 & 0 \end{array} \right) \begin{array}{c} S_1 \\ S_2 \\ S_3 \\ S_4 \\ S_5 \end{array} \end{array}
\tag{36}
$$

$$
p_i = \begin{array}{c} \begin{array}{ccccc} S_1 & S_2 & S_3 & S_4 & S_5 \end{array} \\ \left[ \begin{array}{ccccc} 0 & 4.7 & 2.5 & 0.203 & 3.507 \end{array} \right] \end{array} \text{ppb/hr}
\tag{37}
$$


$$
d_i = \begin{array}{c} \begin{array}{ccccc} S_1 & S_2 & S_3 & S_4 & S_5 \end{array} \\ \left[ \begin{array}{ccccc} 2.1 & 0 & 0 & 4.203 & 0.007 \end{array} \right] \end{array} \text{ppb/hr}
\tag{38}
$$

$$
\tilde{p}_i = \begin{array}{c} \begin{array}{ccccc} S_1 & S_2 & S_3 & S_4 & S_5 \end{array} \\ \left[ \begin{array}{ccccc} 0 & 0.0056 & 0.1476 & 0.0314 & 0.007 \end{array} \right] \end{array} \text{ppb/hr}
\tag{39}
$$

$\quad \tilde{d}_i = \begin{array}{c} \begin{array}{ccccc} S_1 & S_2 & S_3 & S_4 & S_5 \end{array} \\ \left[ \begin{array}{ccccc} 0.1238 & 0 & 0 & 0.0177 & 0.007 \end{array} \right] \end{array} \text{ppb/hr}$
$$
\tag{40}
$$

$$
\tilde{r}_j = \begin{array}{c} \begin{array}{ccccc} R_1 & R_2 & R_3 & R_4 & R_5 \end{array} \\ \left[ \begin{array}{ccccc} 0.081 & 0.064 & 0.067 & 0.067 & 0.04 \end{array} \right] \end{array} \text{ppb/hr}
\tag{41}
$$

### 2.12 Calculation of contributions

We calculate this contribution $C_k$ of a pathway $P_k$ to the production or destruction of a species $S_i$ as the number of molecules
of $S_i$ produced or destroyed by $P_k$ over the number of molecules of $S_i$ produced or destroyed by all pathways:

$$
\begin{aligned}
C_k &= \frac{m_{ik} f_k}{p_i} \quad \text{if } P_k \text{ produces } S_b, \\
C_k &= \frac{m_{ik} f_k}{d_i} \quad \text{if } P_k \text{ destroys } S_b
\end{aligned}
\tag{42}
$$





To calculate the contribution of deleted pathways to the production or destruction of a species we substitute the numerator in equation 42 by $\tilde{p}_i$ and $\tilde{d}_i$ respectively. For example, for the species $HO_2$ in our simple example:

$$C_k = \frac{m_{3k}f_k}{p_3} = \frac{[0,0,1,1,1,1,3][2.704, 0.06, 0.465, 0.111, 0.325, 0.936, 0.172]\text{ppb/hr}}{2.5\text{ppb/hr}}$$

$$= \begin{array}{ccccccc} P_1 & P_2 & P_3 & P_4 & P_5 & P_6 & P_7 \\ \left[ 0 \right. & 0 & 0.186 & 0.044 & 0.13 & 0.374 & \left. 0.206 \right] \end{array} \tag{43}$$

Thus, in this simple example the pathway $P_6$ involving the interaction between reactions $R_3$ and $R_4$ is the most important chain of reactions for the production of $HO_2$, contributing 37.4% of the $HO_2$ production (table 2). The interaction between reactions $R_1$, $R_3$ and $R_4$ (pathway $P_7$) is also important, contributing 20.64% of the $HO_2$ production.

In this simple example, it is easy to see that these reaction chains are important for $HO_2$ production without using the algorithm, but when there are hundreds of reactions interacting the pathway analysis program is a valuable tool to understand
the chemical mechanisms that produce the concentration change of a species in an atmospheric chemistry model. Also, even in this simple example we can see how this algorithm provides valuable quantitative information about the contribution of each pathway to the production of a species.

| ID | Pathway | Contribution (%) | Rate (ppb/hr) |
|---|---|---|---|
| $P_6$ | $H_2O_2 + O \longrightarrow OH + HO_2$<br>$H_2O + HV \longrightarrow OH + O$<br>Net: $H_2O_2 + H_2O \longrightarrow 2\,OH + HO_2$ | 37.43 | 0.936 |
| $P_7$ | $2\,(H_2O_2 + OH \longrightarrow HO_2 + H_2O)$<br>$H_2O_2 + O \longrightarrow OH + HO_2$<br>$H_2O + HV \longrightarrow OH + O$<br>Net: $3\,H_2O_2 \longrightarrow 3\,HO_2 + H_2O$ | 20.64 | 0.172 |
| $P_3$ | $H_2O_2 + OH \longrightarrow HO_2 + H_2O$<br>$H_2O + HV \longrightarrow OH + O$<br>Net: $H_2O_2 \longrightarrow HO_2 + O$ | 18.58 | 0.465 |
| $P_5$ | $OH + OH \longrightarrow H_2O_2$<br>$H_2O_2 + O \longrightarrow OH + HO_2$<br>$H_2O + HV \longrightarrow OH + O$<br>Net: $H_2O \longrightarrow HO_2$ | 13.00 | 0.325 |
| $P_4$ | $H_2O_2 + OH \longrightarrow HO_2 + H_2O$<br>$OH + OH \longrightarrow H_2O_2$<br>$3\,(H_2O + HV \longrightarrow OH + O)$<br>Net: $2\,H_2O \longrightarrow HO_2 + 3\,O$ | 4.42 | 0.111 |





| | | | |
|---|---|---|---|
| Del | Deleted pathways | 5.90 | 0.148 |

Table 2: Contribution of pathways to the production of HO$_2$ in the simple example used to explain the algorithm.

## 3 Implementation

We implement Lehmann's (2004) algorithm in Python. We designed an object-oriented code defining a class to store the
variables listed in table 1 and separating the steps described in section 2 into different class methods. We run these methods in
a main method that performs the loop shown in figure 1. We represent the multiplicities $x_{jk}$ of the pathways as a sparse matrix
to optimize memory usage. Our implementation includes the option to find pathways using multiprocessing to speed up the
computation time.

The code includes several functions that are useful for analyzing the pathways. After the main algorithm loop ends, the
variables in table 1 have all the information of the pathways that have been found. The code includes functions to save these
variables to binary files and to read them for future analysis. The code also has functions to transform the representation of a
pathway from an array of multiplicities to a string, to get the net reaction of a pathway, to create a latex table with the pathways
that contribute to the production or destruction of a species of interest, and to assign a unique identifier to each pathway. This
identifier is a string containing the multiplicities and the indexes of the reactions in a pathway. The same pathway can be
formed multiple times during the algorithm, so we use this identifier to avoid repeating pathways. We also include a function
to calculate the contribution of all pathways to the production or destruction of a species.

Our implementation includes the option to specify species that will be ignored as branching point species. If we are interested
in finding pathways at a specific timescale, it is convenient not to consider species with lifetimes higher than the timescale of
interest as branching-points. Our implementation also includes the option to ignore species as branching-points specifying a
maximum lifetime of branching-point species.

### 3.1 Tests

The code includes run-time tests to ensure that the code works well. If the construction of pathways is correct, the rates of the
reactions must be completely distributed to the pathways:

$$r_j = \tilde{r}_j + \sum_{k=1}^{n_p} x_{jk} f_k, \quad j = 1 \dots n_r \tag{44}$$

This condition ensures that the number of molecules of a species produced or destroyed by the initial reactions is the same as
the number of molecules produced or destroyed by the pathways. The code checks that equation 44 is fulfilled in each iteration
of the algorithm, and displays a warning if it is not satisfied.

The code also includes unit tests to ensure that the code works as expected in a simple scenario with four reactions repre-
senting the Chapman's O$_3$ destruction mechanism (Chapman, 1930). This scenario is used by Lehmann (2004) to explain how





the algorithm works. We include tests to ensure that our implementation finds the same pathways and rates as those found by
       Lehmann (2004) in this very simple example.

## 3.2    Using Chempath

Chempath is available at https://github.com/DanyIvan/chempath. This code repository includes a tutorial on how to use Chempath,
as well as some examples of how to apply Chempath to a box model and to a one-dimensional photochemical model (section
350    4).

There are two important steps to use Chempath. First, the user needs to transform the output of a photochemical model into
files readable by Chempath. The code repository includes some examples of how to create these input files. Second, the user
needs to choose a minimum rate of pathways $f_{\min}$. This can be done by trial and error, or setting it as a fraction of the rate
of production of the species we are interested in finding pathways for. Ideally, $f_{\min}$ will be small enough so that the deleted
pathways do not contribute significantly to the production or destruction of a species of interest. However, if $f_{\min}$ is too small,
the code might take a long time to run.

## 4    Application example: Pathways in a one-dimensional photochemical model

In this section, we show how to apply *Chempath* to the one-dimensional photochemical model *photochem* (Wogan et al. 2023,
https://github.com/Nicholaswogan/photochem). We run the *photochem* model with the "Modern Earth" reaction scheme that
includes 1281 reactions between 113 species. We run the model to photochemical equilibrium using surface flux boundary
conditions for $O_2$, $CH_4$, CO, and $H_2$. We choose fluxes of $3.3 \times 10^{11}, 5 \times 10^{10}, 1.2 \times 10^8$ and $3 \times 10^9$ molecules$/cm^2/s$ for
each of these species respectively. The choice of these fluxes is arbitrary and motivated to get similar conditions to the present
atmosphere. For all other species, we use the default boundary conditions of the "Modern Earth" reaction scheme. After the
model reaches equilibrium, we decrease the $O_2$ surface flux to $2.5 \times 10^{11}$ molecules$/cm^2/s$ and run the model for 5 million
years, getting the output at every time step.

The model output shows that $O_2$ and $O_3$ concentrations tend to decrease at all altitudes as a consequence of the decrease in
the $O_2$ surface input flux (figure 2). It is clear that this $O_3$ concentration decrease is a consequence of the decrease in the $O_2$
surface input flux, but if we want to know what are the chemical mechanisms that contribute to this $O_3$ loss, we need to use
the pathway analysis program. We apply *Chempath* to the *photochem* model output to gain insight into the chemical reaction
chains that destroy $O_3$ in this model run.

### 4.1    Methods: How to find pathways in the *photochem* model

The application of *Chempath* to the *photochem* model output requires the calculation of vertical transport production or de-
struction terms. The *photochem* model calculates the concentration changes of long-lived species solving the equation:



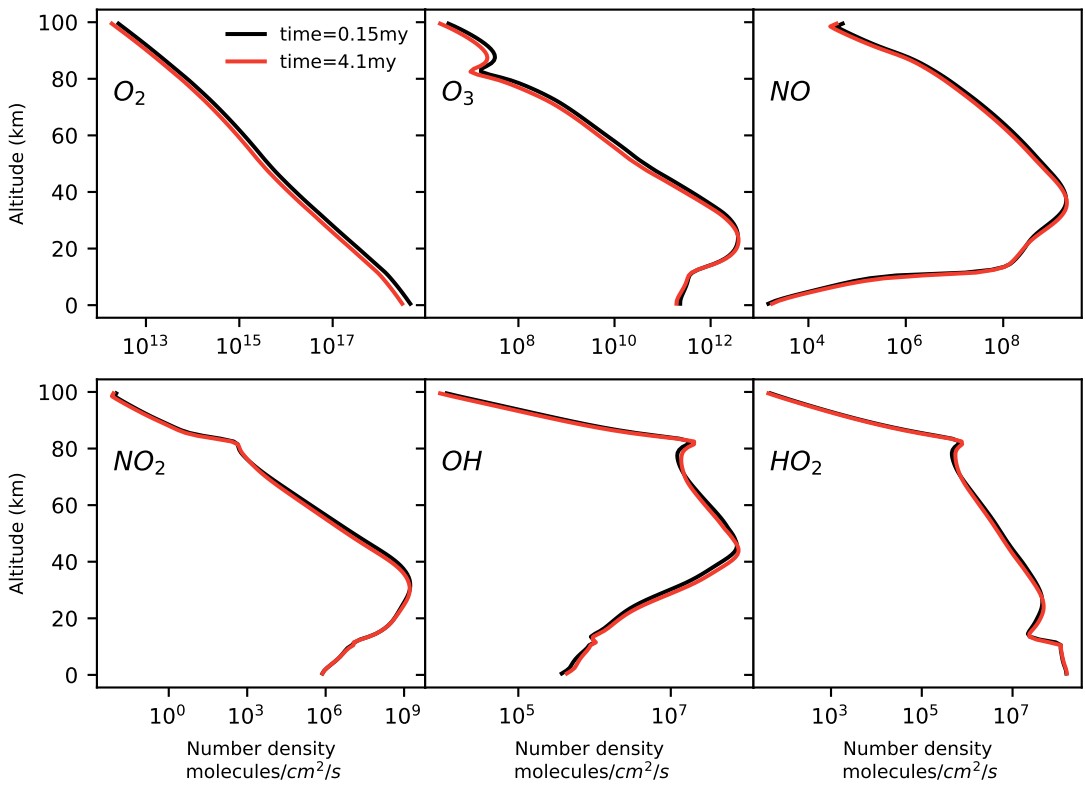

**Figure 2.** Number density profiles of $O_2$, $O_3$, NO, $NO_2$, OH and $HO_2$ calculated by the *photochem* model at time= 0.15 million years (black line) and at time= 4.1 million years (red line). In this model run we decrease the $O_2$ input surface flux. As a result, the $O_2$ and $O_3$ number densities tend to decrease at all altitudes. The concentration of NO, $NO_2$, OH and $HO_2$ decrease and increase at different altitudes.

$$\frac{\partial f_i}{\partial t} = \frac{1}{\rho}\frac{\partial}{\partial z}\Phi_i + \frac{\Pi_i}{\rho} - \frac{L_i}{\rho} - \frac{\Omega_i}{\rho}, \tag{45}$$

where $f_i$ is the mixing ratio of species $i$, $\rho$ is the total number density, $z$ is altitude, $\Pi_i$ and $D_i$ are the chemical production and destruction of species $i$, $\Phi_i$ is the vertical transport flux of species $i$, and $\Omega_i$ is the production or destruction of species $i$ from rainout (see Wogan et al. 2022 for more details). We assume that $\rho$ is constant over time and that equation 45 can be represented as :

$$\frac{d\rho_i}{dt} = \Pi_i - D_i - \Omega_i + T_i, \tag{46}$$

where $\rho_i$ is the number density of species $i$ and $T_i$ is the supply or removal of species $i$ by the vertical transport . We obtain $\Pi_i$, $D_i$, and $\Omega_i$ from the *photochem* model output and use equation 46 to calculate $T_i$ at a given altitude.





To incorporate the effect of vertical transport and rainout into the pathway analysis program we add the following pseudo-reactions to the reaction system for each species $S_i$:

$$S_i \longrightarrow S_{i,\mathrm{trpt}} \text{ if transport supplies } S_i$$
$$S_{i,\mathrm{trpt}} \longrightarrow S_i \text{ if transport removes } S_i \qquad (47)$$
$$S_i \longrightarrow S_{i \text{ rainout}},$$

The rate of these pseudo-reactions is given by the transport rates calculated from equation 46 and the rainout rates calculated by the model.

We run *Chempath* with the updated reaction system at each altitude grid and at 40 time points distributed across the model run. We prescribe a variable minimum pathway rate $f_{\min}$ that we calculate as the minimum of the chemical production of $O_2$, $O_3$, CO, $H_2$ and $CH_4$ divided by 1000. We use these species to calculate $f_{\min}$ because we are interested in understanding their

concentration changes. This $f_{\min}$ choice keeps the contribution of deleted pathways to $O_3$ production and destruction below 5% at all altitudes and times in our analysis. We do not consider these species as branching-points. We also ignore $N_2$, $CO_2$, and $H_2O$ as branching-point species, treating them as long-lived inert species.

### 4.2    Results: Ozone destruction and production pathways in the *photochem* model

*Chempath* allows us to identify the most important pathways for $O_3$ production and destruction at a given altitude and time in

our *photochem* model run (figures 3 and 4, and table 3).

In the troposphere (below 11 km in our model run), $O_3$ production in the *photochem* model is dominated by transport (pathway $P_{2.1}$) and by the photochemical oxidation of hydrocarbons, including methane and the methylperoxy radical (pathways $P_{2.2}$ to $P_{2.4}$) under the presence of nitrogen oxide radicals ($NO_x$). It is widely accepted that tropospheric ozone production is dominated by the $NO_x$ mediated photochemical oxidation of CO and hydrocarbons (see for example Haagen-Smit 1952;

Crutzen 1973; A. Volz-Thomas 1994; Jacob 1999). Our pathway analysis algorithm captures this fact in the *photochem* model output. These hydrocarbon oxidation pathways are similar to the "smog mechanism" that produces tropospheric $O_3$ through oxidation of hydrocarbons (Haagen-Smit, 1952; A. Volz-Thomas, 1994). The tropospheric $O_3$ destruction is dominated by $O_3$ photolysis and subsequent $CH_4$ oxidation under the presence of hydrogen oxide radicals ($HO_x$, pathways $D_{2.1}$ and $D_{2.2}$). Ozone loss catalyzed by $HO_x$ radicals is also important (pathway $D_{2.3}$).

In the stratosphere (11-50km), $O_3$ production is mainly occurring via CO and $CH_4$ oxidation under the presence of $NO_x$ radicals below 25km (pathways $P_{2.6}$ and $P_{2.7}$), and by the Chapman production pathway $P_{2.8}$ above 25km. The main stratospheric $O_3$ destruction mechanisms involve transport (pathway $D_{2.4}$), Chapman-like destruction pathways ($D_{2.5}$, $D_{2.11}$), and destruction by $HO_x$ and $NO_x$ radicals (pathways $D_{2.3}$ and $D_{2.6}$ to $D_{2.10}$). Catalytic cycles involving $NO_x$ and $HO_x$ radicals are important for stratospheric $O_3$ destruction (Lary, 1997; Jacob, 1999). Our analysis identifies these important catalytic cycles.





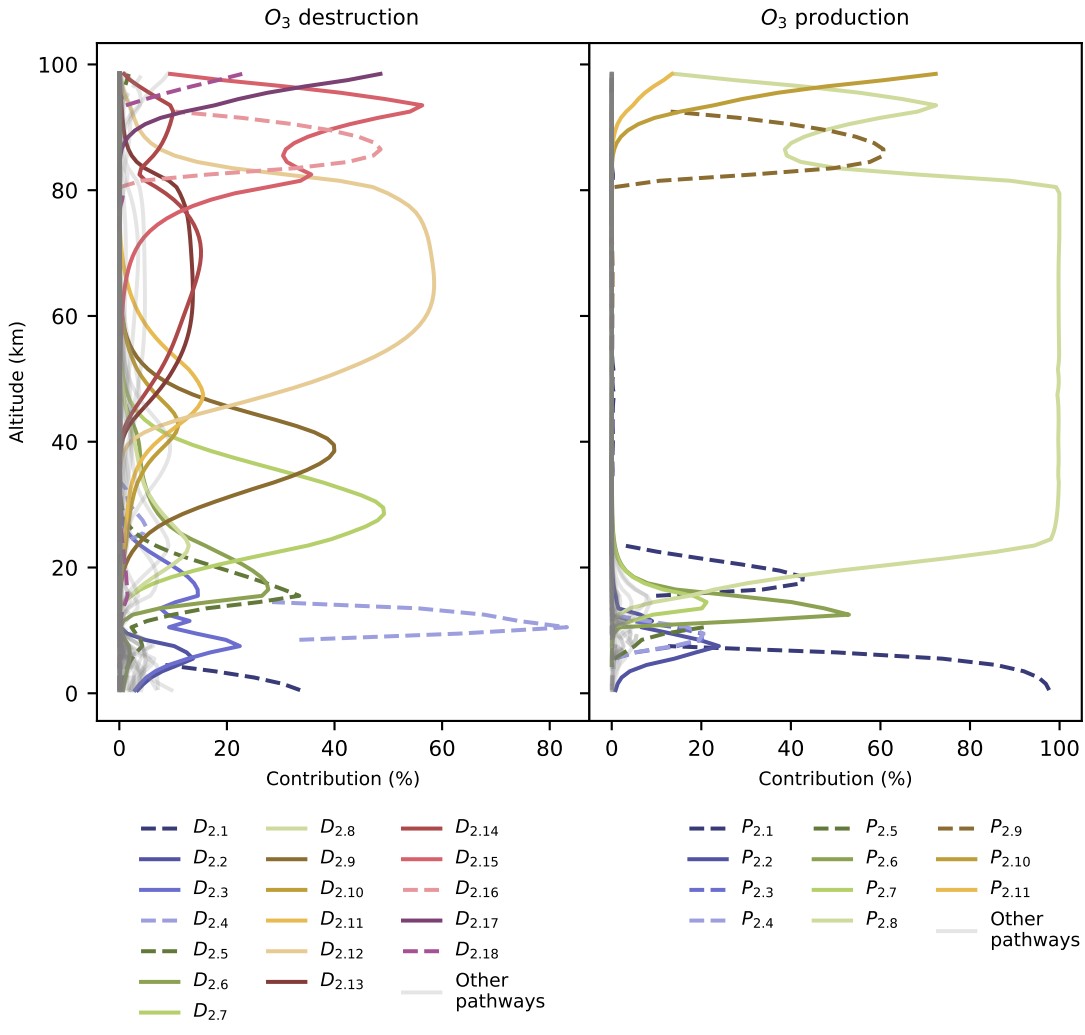

**Figure 3.** Contribution of pathways to O₃ production and destruction as a function of altitude at the model time= 4.1 million years. The main pathways are plotted in color and the less important pathways are plotted in gray. Pathways that include transport reactions are plotted in dashed lines. Pathways that contain transport pseudo-reactions show discontinuities because transport can either supply or remove $O_3$ at different altitudes. The pathways are listed in table 3.



Above 50km of altitude, the main $O_3$ production mechanisms are Chapman-like production pathways ($P_{2.8}$ to $P_{2.11}$), and the main $O_3$ destruction mechanisms involve $O_3$ photolysis coupled to $HO_x$ radicals cycles (pathways $D_{2.12}$ and $D_{2.13}$), destruction by $HO_x$ radicals (pathways $D_{2.14}$ to $D_{2.17}$), and Chapman-like destruction pathways ($P_{2.18}$).

The contribution profiles shown in figure 3 have a similar structure in all the time steps we analyzed. For example, at 30km of altitude, the pathways producing and destroying $O_3$ have very similar contributions over time (figure 4). Consequently, the

pathways shown in figure 3 and listed in table 3 are a good representation of the pathways that produce and destroy $O_3$ and $O_2$ across all times in the model run. These pathways are similar to those found in a previous study of chemical pathways affecting $O_3$ in the atmosphere (Grenfell et al., 2006).

Our analysis suggests that the decrease in $O_3$ shown in figure 2 is caused by destruction pathways involving $NO_x$ and $HO_x$ radicals and Chapman reactions. The fact that *Chempath* found the presence of these well-known pathways in a photochemical

model of Earth's atmosphere suggests that this algorithm can be applied to understand the chemistry of less well-characterized atmospheres, like exoplanet or past atmospheres.

| ID | Pathway | Maximum contribution % | Rate $\frac{molecules}{cm^2 s}$ | Altitude of maximum contribution km |
|---|---|---|---|---|
| $D_{2.1}$ | $2\,(CH_4 + OH \longrightarrow CH_3 + H_2O)$<br>$O_1D + H_2O \longrightarrow OH + OH$<br>$2\,(CH_3 + O_2 + M \longrightarrow CH_3O_2 + M)$<br>$O_3 + hv \longrightarrow O_1D + O_2$<br>$2\,(CH_3O_2 \longrightarrow CH_3O_{2trpt})$<br>Net: $O_2 + 2\,CH_4 + O_3 \longrightarrow H_2O + 2\,CH_3O_{2trpt}$ | 33.608 | 2.925e+04 | 0.5 |
| $D_{2.2}$ | $2\,(CH_4 + OH \longrightarrow CH_3 + H_2O)$<br>$2\,(CH_3O + O_2 \longrightarrow H_2CO + HO_2)$<br>$HO_2 + HO_2 \longrightarrow H_2O_2 + O_2$<br>$2\,(CH_3 + O_2 + M \longrightarrow CH_3O_2 + M)$<br>$2\,(CH_3O_2 + O \longrightarrow CH_3O + O_2)$<br>$2\,(H_2CO + hv \longrightarrow CO + H_2)$<br>$2\,(O_3 + hv \longrightarrow O + O_2)$<br>$H_2O_2 + hv \longrightarrow OH + OH$<br>Net: $2\,CH_4 + 2\,O_3 \longrightarrow 2\,H_2 + 2\,H_2O + O_2 + 2\,CO$ | 13.697 | 2.866e+03 | 5.5 |



| $D_{2.3}$ | $HO_2 + HO_2 \longrightarrow H_2O_2 + O_2$<br>$2\,(OH + O_3 \longrightarrow HO_2 + O_2)$<br>$H_2O_2 + h\nu \longrightarrow OH + OH$<br>Net: $2\,O_3 \longrightarrow 3\,O_2$ | 22.404 | 3.210e+03 | 7.5 |
| $D_{2.4}$ | $O_3 \longrightarrow O_{3\,trpt}$<br>Net: $O_3 \longrightarrow O_{3\,trpt}$ | 83.208 | 5.948e+04 | 10.5 |
| $D_{2.5}$ | $O_3 + h\nu \longrightarrow O + O_2$<br>$O \longrightarrow O_{trpt}$<br>Net: $O_3 \longrightarrow O_2 + O_{trpt}$ | 33.453 | 8.377e+03 | 15.5 |
| $D_{2.6}$ | $O_1D + H_2O \longrightarrow OH + OH$<br>$OH + HO_2 \longrightarrow H_2O + O_2$<br>$OH + O_3 \longrightarrow HO_2 + O_2$<br>$O_3 + h\nu \longrightarrow O_1D + O_2$<br>Net: $2\,O_3 \longrightarrow 3\,O_2$ | 27.694 | 5.007e+03 | 16.5 |
| $D_{2.7}$ | $NO + O_3 \longrightarrow NO_2 + O_2$<br>$NO_2 + O \longrightarrow NO + O_2$<br>$O_3 + h\nu \longrightarrow O + O_2$<br>Net: $2\,O_3 \longrightarrow 3\,O_2$ | 49.204 | 3.714e+05 | 28.5 |
| $D_{2.8}$ | $HO_2 + O \longrightarrow OH + O_2$<br>$OH + O_3 \longrightarrow HO_2 + O_2$<br>$O_3 + h\nu \longrightarrow O + O_2$<br>Net: $2\,O_3 \longrightarrow 3\,O_2$ | 12.92 | 1.980e+04 | 23.5 |
| $D_{2.9}$ | $O_1D + N_2 \longrightarrow O + N_2$<br>$NO + O_3 \longrightarrow NO_2 + O_2$<br>$NO_2 + O \longrightarrow NO + O_2$<br>$O_3 + h\nu \longrightarrow O_1D + O_2$<br>Net: $2\,O_3 \longrightarrow 3\,O_2$ | 39.97 | 9.662e+05 | 38.5 |
| $D_{2.10}$ | $O_1D + N_2 \longrightarrow O + N_2$<br>$HO_2 + O \longrightarrow OH + O_2$<br>$OH + O_3 \longrightarrow HO_2 + O_2$<br>$O_3 + h\nu \longrightarrow O_1D + O_2$<br>Net: $2\,O_3 \longrightarrow 3\,O_2$ | 10.943 | 2.328e+05 | 42.5 |



| | | | | |
|---|---|---|---|---|
| $D_{2.11}$ | $O_1D + N_2 \longrightarrow O + N_2$<br>$O + O_3 \longrightarrow O_2 + O_2$<br>$O_3 + hv \longrightarrow O_1D + O_2$<br>Net: $2O_3 \longrightarrow 3O_2$ | 15.596 | 2.307e+05 | 47.5 |
| $D_{2.12}$ | $O + OH \longrightarrow O_2 + H$<br>$2(O_1D + N_2 \longrightarrow O + N_2)$<br>$H + O_2 + M \longrightarrow HO_2 + M$<br>$HO_2 + O \longrightarrow OH + O_2$<br>$2(O_3 + hv \longrightarrow O_1D + O_2)$<br>Net: $2O_3 \longrightarrow 3O_2$ | 58.521 | 2.554e+05 | 65.5 |
| $D_{2.13}$ | $O + OH \longrightarrow O_2 + H$<br>$2(O_1D + O_2 \longrightarrow O + O_2)$<br>$H + O_2 + M \longrightarrow HO_2 + M$<br>$HO_2 + O \longrightarrow OH + O_2$<br>$2(O_3 + hv \longrightarrow O_1D + O_2)$<br>Net: $2O_3 \longrightarrow 3O_2$ | 13.672 | 6.625e+04 | 63.5 |
| $D_{2.14}$ | $O + OH \longrightarrow O_2 + H$<br>$O_1D + N_2 \longrightarrow O + N_2$<br>$H + O_3 \longrightarrow OH + O_2$<br>$O_3 + hv \longrightarrow O_1D + O_2$<br>Net: $2O_3 \longrightarrow 3O_2$ | 15.18 | 4.997e+04 | 70.5 |
| $D_{2.15}$ | $2(O + OH \longrightarrow O_2 + H)$<br>$2(H + O_3 \longrightarrow OH + O_2)$<br>$O_2 + hv \longrightarrow O + O$<br>Net: $2O_3 \longrightarrow 3O_2$ | 56.315 | 5.890e+04 | 93.5 |
| $D_{2.16}$ | $O + OH \longrightarrow O_2 + H$<br>$H + O_3 \longrightarrow OH + O_2$<br>$O_{trpt} \longrightarrow O$<br>Net: $O_3 + O_{trpt} \longrightarrow 2O_2$ | 48.573 | 2.281e+05 | 86.5 |
| $D_{2.17}$ | $2(O + OH \longrightarrow O_2 + H)$<br>$O_1D + N_2 \longrightarrow O + N_2$<br>$2(H + O_3 \longrightarrow OH + O_2)$<br>$O_2 + hv \longrightarrow O + O_1D$<br>Net: $2O_3 \longrightarrow 3O_2$ | 48.536 | 1.082e+04 | 98.5 |



| | | | | |
|---|---|---|---|---|
| $D_{2.18}$ | $O_1D + N_2 \longrightarrow O + N_2$<br>$O_3 + hv \longrightarrow O_1D + O_2$<br>$O \longrightarrow O_{trpt}$<br>Net: $O_3 \longrightarrow O_2 + O_{trpt}$ | 22.849 | 1.019e+04 | 98.5 |
| $P_{2.1}$ | $O_{3trpt} \longrightarrow O_3$<br>Net: $O_{3trpt} \longrightarrow O_3$ | 97.677 | 8.501e+04 | 0.5 |
| $P_{2.2}$ | $2(CH_4 + OH \longrightarrow CH_3 + H_2O)$<br>$2(CH_3O + O_2 \longrightarrow H_2CO + HO_2)$<br>$HO_2 + HO_2 \longrightarrow H_2O_2 + O_2$<br>$2(O + O_2 + M \longrightarrow O_3 + M)$<br>$2(CH_3 + O_2 + M \longrightarrow CH_3O_2 + M)$<br>$2(CH_3O_2 + NO \longrightarrow CH_3O + NO_2)$<br>$2(H_2CO + hv \longrightarrow CO + H_2)$<br>$H_2O_2 + hv \longrightarrow OH + OH$<br>$2(NO_2 + hv \longrightarrow NO + O)$<br>Net: $5O_2 + 2CH_4 \longrightarrow 2H_2 + 2H_2O + 2CO + 2O_3$ | 23.943 | 3.430e+03 | 7.5 |
| $P_{2.3}$ | $4(CH_3O + O_2 \longrightarrow H_2CO + HO_2)$<br>$3(HO_2 + HO_2 \longrightarrow H_2O_2 + O_2)$<br>$2(H_2O_2 + OH \longrightarrow HO_2 + H_2O)$<br>$4(O + O_2 + M \longrightarrow O_3 + M)$<br>$4(CH_3O_2 + NO \longrightarrow CH_3O + NO_2)$<br>$4(H_2CO + hv \longrightarrow CO + H_2)$<br>$H_2O_2 + hv \longrightarrow OH + OH$<br>$4(NO_2 + hv \longrightarrow NO + O)$<br>$4(CH_3O_{2trpt} \longrightarrow CH_3O_2)$<br>Net: $5O_2 + 4CH_3O_{2trpt} \longrightarrow 4H_2 + 2H_2O + 4CO + 4O_3$ | 20.146 | 1.726e+03 | 8.5 |





| | | | | |
|---|---|---|---|---|
| $P_{2.4}$ | $2\,(OH + HO_2 \longrightarrow H_2O + O_2)$<br>$4\,(CH_3O + O_2 \longrightarrow H_2CO + HO_2)$<br>$HO_2 + HO_2 \longrightarrow H_2O_2 + O_2$<br>$4\,(O + O_2 + M \longrightarrow O_3 + M)$<br>$4\,(CH_3O_2 + NO \longrightarrow CH_3O + NO_2)$<br>$4\,(H_2CO + h\nu \longrightarrow CO + H_2)$<br>$H_2O_2 + h\nu \longrightarrow OH + OH$<br>$4\,(NO_2 + h\nu \longrightarrow NO + O)$<br>$4\,(CH_3O_{2trpt} \longrightarrow CH_3O_2)$<br>$Net: 5\,O_2 + 4\,CH_3O_{2trpt} \longrightarrow 4\,H_2 + 2\,H_2O + 4\,CO + 4\,O_3$ | 20.519 | 2.457e+03 | 9.5 |
| $P_{2.5}$ | $2\,(CH_3O + O_2 \longrightarrow H_2CO + HO_2)$<br>$HO_2 + HO_2 \longrightarrow H_2O_2 + O_2$<br>$2\,(O + O_2 + M \longrightarrow O_3 + M)$<br>$2\,(CH_3O_2 + NO \longrightarrow CH_3O + NO_2)$<br>$2\,(H_2CO + h\nu \longrightarrow CO + H_2)$<br>$2\,(NO_2 + h\nu \longrightarrow NO + O)$<br>$H_2O_2 \longrightarrow H_2O_{2trpt}$<br>$2\,(CH_3O_{2trpt} \longrightarrow CH_3O_2)$<br>$Net: 3\,O_2 + 2\,CH_3O_{2trpt} \longrightarrow 2\,H_2 + 2\,CO + 2\,O_3 + H_2O_{2trpt}$ | 20.548 | 7.345e+03 | 10.5 |
| $P_{2.6}$ | $CO + OH \longrightarrow CO_2 + H$<br>$H + O_2 + M \longrightarrow HO_2 + M$<br>$O + O_2 + M \longrightarrow O_3 + M$<br>$NO + HO_2 \longrightarrow NO_2 + OH$<br>$NO_2 + h\nu \longrightarrow NO + O$<br>$Net: 2\,O_2 + CO \longrightarrow CO_2 + O_3$ | 52.975 | 1.379e+04 | 12.5 |
| $P_{2.7}$ | $CH_4 + OH \longrightarrow CH_3 + H_2O$<br>$CH_3O + O_2 \longrightarrow H_2CO + HO_2$<br>$2\,(O + O_2 + M \longrightarrow O_3 + M)$<br>$NO + HO_2 \longrightarrow NO_2 + OH$<br>$CH_3 + O_2 + M \longrightarrow CH_3O_2 + M$<br>$CH_3O_2 + NO \longrightarrow CH_3O + NO_2$<br>$H_2CO + h\nu \longrightarrow CO + H_2$<br>$2\,(NO_2 + h\nu \longrightarrow NO + O)$<br>$Net: 4\,O_2 + CH_4 \longrightarrow H_2 + H_2O + CO + 2\,O_3$ | 21.181 | 2.425e+03 | 14.5 |





| $P_{2.8}$ | $2\,(O + O_2 + M \longrightarrow O_3 + M)$ <br> $O_2 + h\nu \longrightarrow O + O$ <br> Net: $3\,O_2 \longrightarrow 2\,O_3$ | 99.959 | 8.865e+04 | 79.5 |
|---|---|---|---|---|
| $P_{2.9}$ | $O + O_2 + M \longrightarrow O_3 + M$ <br> $O_{\text{trpt}} \longrightarrow O$ <br> Net: $O_2 + O_{\text{trpt}} \longrightarrow O_3$ | 60.679 | 2.850e+05 | 86.5 |
| $P_{2.10}$ | $O_1D + N_2 \longrightarrow O + N_2$ <br> $2\,(O + O_2 + M \longrightarrow O_3 + M)$ <br> $O_2 + h\nu \longrightarrow O + O_1D$ <br> Net: $3\,O_2 \longrightarrow 2\,O_3$ | 72.342 | 1.613e+04 | 98.5 |
| $P_{2.11}$ | $O_1D + O_2 \longrightarrow O + O_2$ <br> $2\,(O + O_2 + M \longrightarrow O_3 + M)$ <br> $O_2 + h\nu \longrightarrow O + O_1D$ <br> Net: $3\,O_2 \longrightarrow 2\,O_3$ | 13.452 | 3.000e+03 | 98.5 |

Table 3: Pathways producing and destroying $O_3$ at time= 4.5 million years of the model run. The contribution profiles of these pathways are shown in figure 3. The contributions and rates in this table correspond to the height at which the pathways contribute the most to $O_3$ production and destruction.



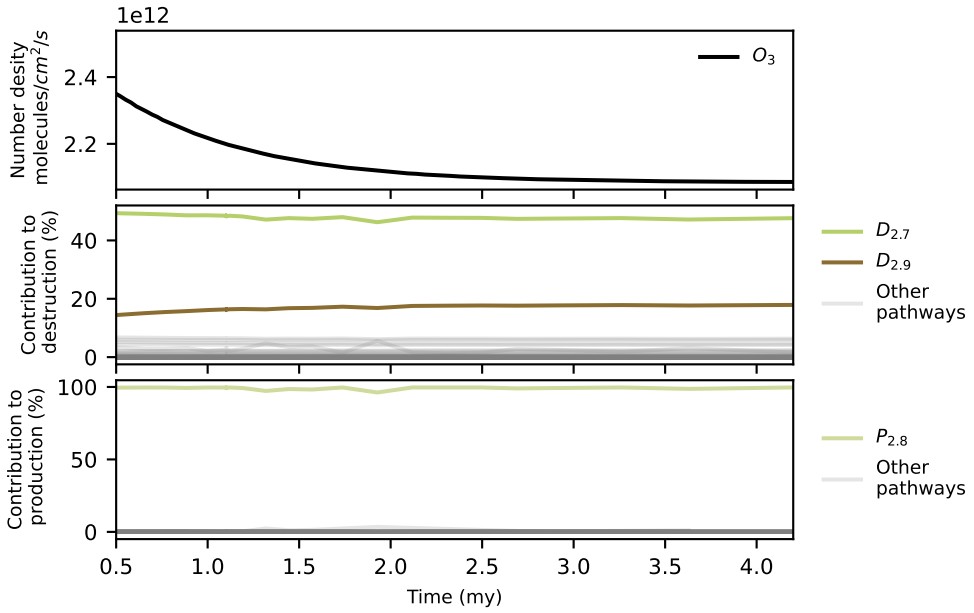

**Figure 4.** Contributions of the main pathways producing (middle panel) and destroying (bottom panel) $O_3$ as a function of time at a 30km altitude. The pathways are listed in tables 3. The time evolution of the $O_3$ number density at 30km is shown in the top panel.

## 5 Conclusions

In this paper, we described the development of *Chempath*: an open-source pathway analysis program for photochemical models that can automatically construct the most relevant pathways of a reaction system and identify the most important pathways for

the production and destruction of a species of interest. We showed how to use *Chempath* in a one-dimensional photochemical model. *Chempath* identified well-known pathways for $O_3$ destruction and production in Earth's atmosphere, suggesting that this algorithm can be used to understand chemical mechanisms in photochemical models of less well-known atmospheres, like exoplanet or past atmospheres.

*Code availability.* A frozen version of the code used in this paper is available at https://doi.org/10.5281/zenodo.13715328. For up-to-date

developments see the *Chempath* GitHub repository: https://github.com/DanyIvan/chempath. This repository includes Jupyter notebooks that describe how to run and reproduce the examples presented in this paper.

*Author contributions.* DGR: Conceptualization, Software, Investigation, Writing - original draft preparation. CG: Supervision, Funding acquisition, Writing - review & editing. ASA: Supervision, Funding acquisition, Writing - review & editing.





*Competing interests.* We declare that none of the authors has any competing interests.

*Acknowledgements.* We acknowledge and respect the ləkʷʼəŋən peoples on whose traditional territory the university of Victoria stands and the Songhees, Esquimalt and W̱SÁNEĆ peoples whose historical relationships with the land continue to this day. We thank Ralph Lehmann for answering questions about the pathway analysis program. Primary financial support came from Natural Science and Engineering Research Council of Canada (NSERC) Discovery Grants to Colin Goldblatt (RGPIN-2018-05929) and Anne-Sofie Ahm (RGPIN-2022-03912). High-performance computing facilities were provided via a NSERC Research Tools and Infrastructure Grant (RTI-2020-00277).



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
