# Peer review of "Chempath 1.0: An open-source pathway analysis program for photochemical models"

_Geoscientific Model Development, 2024_

## Referee Comment (RC1)

**Review of**

**"Chempath 1.0: An open-source pathway analysis program for photochemical models" by D. Garduño Ruiz et al.**

(Numbers refer to line numbers.)

**General**

The authors describe an implementation of the (existing) pathway analysis program of Lehmann (2004) in Python. This may please potential users who prefer to run a code in the younger programming language Python over Fortran.

**Questions**

- Before the actual determination of pathways, the original algorithm checks the balance of the input data (= output of a chemical model) (Lehmann, 2002, Section 3.1): Are the reaction rates consistent with the concentration changes calculated by the model? This is an essential step to detect (and possibly correct) imbalances, which may arise from numerical inaccuracies (e.g., because of a large time step in the chemical model). Balanced input data are indispensable for the actual pathway formation.
The authors do not mention how they solve this problem in their implementation.

221-222 How does the algorithm choose a splitting of a pathway into subpathways if this splitting is not unique?
Does the formulation "we choose the solution that minimizes the most equation 19" mean that a solution of

$$\min \left\{ \frac{1}{2} \cdot ||Ax - b||^2 \mid x \geq 0 \right\} \qquad \qquad \hat{=} (19)$$

is chosen? Although it is true that the optimization problem (19) has a global minimum (line 221), it is not guaranteed that there is a unique solution. In fact, if Equation (17) has multiple solutions, this will also be the case for the related optimization problem (19).

- Table 3: Pathways $D_{2,1}$, $P_{2,3}$, $P_{2,4}$, $P_{2,5}$: In the troposphere $CH_3O_2$ has a short chemical lifetime (usually $< 1$ min). Therefore we would expect that its "fate" is controlled by local chemistry, not by transport. Nevertheless, the pathways mentioned above involve transport of $CH_3O_2$. Is it possible that this transport of $CH_3O_2$ is a numerical artefact, resulting from its calculation according to Eq. (46)? There the (possibly small) contribution of transport $T_i$ is calculated as the difference of (possibly larger) chemical terms.

Pathway $D_{2,5}$: The same type of question applies to $D_{2,5}$: This pathway involves transport of atomic oxygen, which has a short chemical lifetime in the altitude region indicated (around 15.5 km).

**Details**

As mentioned above, the authors describe a re-implementation of the (existing) pathway analysis program of Lehmann (2004) in Python. This is said correctly in the body of the text, but it would be good to state this more clearly in the abstract (and, maybe, conclusions) for the "hasty" reader.

36-37 This leads to a difficult discussion. "Available on request" may also be considered as a form of "open" - with the additional advantage that through the personal contact the user can obtain all support needed. Anonymous download of a program is risky if the user does not understand perfectly the functionality and limitations of the program, which may be hard to achieve even if there is a good documentation. I understand that the authors want to provide a reason for their re-programming effort, but I think that the formulation in line 38 is sufficient for that.

64-72 Strictly speaking, "ppb" is the unit for mixing ratio, not concentration.

- Table 1: Row "$s_{ij}$": For readers familiar with chemical systems you might add "stoichiometric matrix".

- Table 1: Rows "$\tilde{r}_j$, $\tilde{p}_i$, $\tilde{d}_i$": The reader might be surprised by these definitions, because deleted pathways have not been mentioned before. An earlier mentioning of deleted pathways would also be beneficial for Section 2.5 (describing the calculation of rates of deleted pathways), which is placed before Section 2.7, where deleted pathways are introduced.

(and several other lines) The official symbol four "hour" is "h". Please insert blanks between numbers and units.

The notation $\sum_{j=1}^{5}[1, 0, 1, 0, 0] \cdot [1, 05, 1.5, 5, 0.1]$ does not make sense, since there is no "$j$" in the terms after the sum. A workaround might consist in writing the sum explicitly ("$1 \cdot 1 + 0 \cdot 0.5 + ...$") or as scalar product.

I would not mention "the reaction system" together with "in two consecutive time steps" (it the same for both time steps).

"mean reaction rates": "in two consecutive time steps" or "between two consecutive time steps" or "within one time step"?

It may happen that the concentration of a species at two points in time $t_1$ and $t_1 + \Delta t$ is not sufficient as input, but information on the concentration between $t_1$ and $t_1 + \Delta t$, e.g. its mean, is also needed: For instance, if you analyse tropospheric chemistry over a full day (from midnight $t_1$ to the next midnight $t_1 + \Delta t$, $\Delta t = 24$ h), you will obtain $[OH] \approx$ zero at $t_1$ and $t_1 + \Delta t$ (although OH is present during the day), which leads to a wrong estimate of the lifetime of OH in $[t_1, t_1 + \Delta t]$ (needed in line 94).

$f_j = r_j$ instead of $f_{kinit} = r_j$?

- Fig. 1: Box 8: "Recomputation of variables": Which variables? How?

"... produced (or consumed) by one pathway is consumed (or produced) by another pathway" might be a bit clearer than the present formulation.

123-124 As these operations are carried out for each pathway separately (i.e. $g$ may differ from pathway to pathway), the sentence should be formulated in singular: "The multiplicities $x_{ij}$ of a new pathway ...The rate of the new pathway is multiplied..."

125, 130 (and several other lines) Throughout the manuscript the authors use identical denotations for elements of a matrix (or vector) and the whole matrix (or vector). Although the reader may "guess" what is meant, I recommend a stricter notation, especially since the number of indices does not always indicate the dimension of the object, e.g. "$x_{jn}$" in line 130 denotes a vector.

134, 197, 256, 287: Why "similar to" instead of "has the following form"?

"in the previous step" $\Rightarrow$ "from the previous step" (or omit completely)?

Eq. (10): "$\sum_k$" $\Rightarrow$ "$\sum_e$" (3 times)

179-181 Lines 176-181 describe the deletion of pathways that have been "used" (i.e. connected to other pathways). This step does not involve any consideration of deletion due to rates $< f_{min}$. Why are $\tilde{r}_j, \tilde{p}_i, \tilde{d}_i$ mentioned nevertheless? (And what exactly does "this case" refer to?)

"minimizing the equation" $\Rightarrow$ "minimizing the expression"

"$x \leq \infty$" $\Rightarrow$ "$x < \infty$"

A comparison with the variables in (17) indicates that:

$$A = ((x'_{je}))_{j=1,\ldots,n_r,\,e=1,\ldots,n_e}$$
$$b = (x_{jc})_{j=1,\ldots,n_r}$$

$0.007 \Rightarrow 0.0073$ (in order to be consistent with Eq. (30))

"this contribution" $\Rightarrow$ "the contribution"

Eq. (42): $S_b \Rightarrow S_i$ (2 times)

308-309 It seems that "For example, ... (43)" should be placed directly after Eq. (42), not after text about deleted pathways.

Eq. (43): This equation involves element-wise multiplication of two vectors (not scalar product). Perhaps this should be said explicitly.

   - Table 2: "HV" $\Rightarrow$ "$h\nu$" (several times)

353-354 "setting it as a fraction of the rate of production of the species...": Here "production" refers to the total production by all reactions? If so, you might emphasize this, in order to avoid confusion with the production by pathways in Table 3. In general, this way of choosing $f_{min}$ may still require further "trial and error": If the species of interest ($S_i$) is involved in zero cycles with large rates (e.g. $O_3 \leftrightarrow O + O_2$), then the rate of the (total) production of $S_i$ will be much larger than the net production or destruction, which shall be explained by pathways. This may have the consequence that an originally chosen fraction (i.e. $f_{min}$) may turn out to be too large and must be reduced.

Shouldn't destruction rates also be taken into account, e.g. for species like $CH_4$ that is only destroyed in Earth's atmosphere?

How many of the 1281 reactions have a rate $> f_{min}$, so that they actually take part in the formation of pathways?

Please explain to the reader the idea behind the reduction of the $O_2$ surface flux.
Which processes in the model lead to a removal of $O_2$ (eventually balancing the source by the surface flux)? What is the time scale of these processes?

"every time step": How long is one time step?

368-370 "... if we want to know what are the chemical mechanisms that contribute to this $O_3$ loss, we need to use the pathway analysis program. We apply *Chempath* to the *photochem* model output to gain insight into the chemical reaction chains that destroy $O_3$ in this model run.":

This formulation sounds as if the $O_3$ destruction pathways directly explain the $O_3$ decrease (probably of a few ppb / million years $\sim 10^{-6}$ ppb/y) occurring in the model run after the reduction of the $O_2$ surface flux. However, this is not the case. As $O_x$ $(= O_3 + O + O(^1D))$ has a chemical lifetime of $\leq 1$ year below 100 km (and much less in the middle atmosphere) (e.g., G. Brasseur and S. Solomon: Aeronomy of the Middle Atmosphere, Springer, Dordrecht, 2005: Fig. 5.3), it will be close to equilibrium in your million-year long model run, i.e. the concentration is determined by the production rate (strongly dependent on the changing $[O_2]$) and the time scale of destruction. By the way, these arguments may serve as motivation for showing production and destruction pathways later on (Table 3).

372-373 "vertical transport production and destruction" $\Rightarrow$ e.g., "supply and removal by vertical transport"?

- Figure 2 (upper left panel): "m" means milli $= 10^{-3}$.

Eq. (45): "$L_i \Rightarrow$ "$D_i$"

Does "production by rainout" mean that evaporation of rain and release of trace gases from the liquid phase to the gas phase is included in the model? According to Eq. (47) this seems not to be the case.

You might include $\frac{d\rho_i}{dt}$ in the list of values obtained from the model.

Eq. (47): It seems that "supplies" and "removes" should be interchanged.

388-389 $CH_4$ is not photochemically produced in Earth's atmosphere (G. Brasseur and S. Solomon: Aeronomy of the Middle Atmosphere, Springer, Dordrecht, 2005: p. 296). If this is true also in your model, then $f_{min}$ as defined in lines 388-389 will be zero. Please clarify.

- Figure 3: It would be nice to use similar colours (or additional symbols) to indicate pathways of the same "family" ($HO_x$, $NO_x$ etc.).

- Figure 3, Table 3: It might be more logical to present production pathways before destruction pathways.

- Figure 3, Table 3 and related text: All pathways have the same first index 2. Therefore it might be omitted in the manuscript (probably it results from the fact that $O_3$ is species no. 2 in the model).

- Table 3: "$O_1D$" $\Rightarrow$ "$O(^1D)$" (several times)

- Table 3: It seems that the algorithm does not attempt to order the reactions in a way that the flux of molecules can be easily followed by the user. I suggest to do that "by hand" (and clearly state that it was done in this way); e.g. pathway $D_{(2,)2}$ might be reordered in the following way:

$$\; (O_3 + hv \longrightarrow O + O_2 )$$
$$\; (CH_4 + OH \longrightarrow CH_3 + H_2 O)$$
$$\; (CH_3 + O_2 + M \longrightarrow CH_3O_2 + M)$$
$$\; (CH_3O_2 + O \longrightarrow CH_3O + O_2 )$$
$$\; (CH_3O + O_2 \longrightarrow H_2CO + HO_2 )$$
$$\; (H_2CO + hv \longrightarrow CO + H_2 )$$
$$HO_2 + HO_2 \longrightarrow H_2O_2 + O_2$$
$$H_2O_2 + h\nu \longrightarrow OH + OH$$

- Figure 4: Upper panel: Delete "s" in the unit of the number density

- Figure 4: Caption: "middle panel" and "bottom panel" does not coincide with the figure above.

- Figure 4: Caption: "tables 3" $\Rightarrow$ "table 3"

---

## Author Comment (AC1)

**Responses to questions in Review**

Daniel Garduno Ruiz, Colin Goldblatt, Anne-Sofie Ahm

October 29, 2024

Thank you so much for a thorough and detailed review. We will address all the comments in our final response, which will be submitted after the open discussion ends (after the 6th of November). Here we respond to major questions in the review:

1. Before the actual determination of pathways, the original algorithm checks the balance of the input data (= output of a chemical model) (Lehmann, 2002, Section 3.1): Are the reaction rates consistent with the concentration changes calculated by the model? This is an essential step to detect (and possibly correct) imbalances, which may arise from numerical inaccuracies (e.g., because of a large time step in the chemical model). Balanced input data are indispensable for the actual pathway formation. The authors do not mention how they solve this problem in their implementation.

   **Response**: We assume that the user ensures mass balance is fulfilled in their model output. Our implementation does not try to correct imbalances. However, in response to this comment we updated the code to include a function that displays a warning if any of the species are not balanced by the reactions. The warning is displayed if balance is not fulfilled to a relative tolerance of $1 \times 10^{-3}$ if the concentration change is greater or equal to 1 molecule/$cm^3$, and to an absolute tolerance of $1 \times 10^{-3}$ if the concentration change is lower than 1 molecule/$cm^3$. We use an absolute tolerance for concentration changes lower than 1 molecule/$cm^3$ because we consider that concentration changes lower than $1 \times 10^{-3}$ molecules/$cm^3$ are unimportant. We will update the text to clarify the necessity of mass balance in the chemical model to be analyzed.

   In the example presented in section 4 of the manuscript, the application of equation 46 ensures that the concentration changes are balanced by the reaction, transport, and rainout rates. In our analysis, the function described above does not display a warning. For many species the balance has a better precision than a relative tolerance of $10^{-3}$, but for some species like O the precision is not better than $10^{-4}$. That is why we chose a relative tolerance of $10^{-3}$ for our warning function.

2. How does the algorithm choose a splitting of a pathway into subpathways if this splitting is not unique? Does the formulation "we choose the solution that minimizes the most equation 19" mean that a solution of equation 19 is chosen? Although it is true that the optimization problem (19) has a global minimum (line 221), it is not guaranteed that there is a unique solution. In fact, if Equation (17) has multiple solutions, this will also be the case for the related optimization problem (19).

**Response**: We chose the first solution that minimizes equation 19 found by Scipy's lsq_linear algorithm. As a response to this comment, we updated the code to include the option to solve equation 17 of the manuscript with the procedure described in section 5.5.1 of Lehmann (2004). We show a comparison of $O_3$ destruction pathways using these two different methods of solving equation 17 in figure 1 and table 1. Our results are very similar using the two methods of solving equation 17. We will update the text to describe this new option.

| ID | Pathway | Contribution lsq_linear % | Contribution Lehmann (2004) % | Alt km |
|---|---|---|---|---|
| $D_{2.1}$ | $2\,(CH_4 + OH \longrightarrow CH_3 + H_2O)$
 $O(^1D) + H_2O \longrightarrow OH + OH$
 $2\,(CH_3 + O_2 + M \longrightarrow CH_3O_2 + M)$
 $O_3 + hv \longrightarrow O(^1D) + O_2$
 $2\,(CH_3O_2 \longrightarrow CH_3O_{2trpt})$
 Net: $O_2 + 2\,CH_4 + O_3 \longrightarrow H_2O + 2\,CH_3O_{2trpt}$ | 33.608 | 33.611 | 0.5 |
| $D_{2.2}$ | $2\,(CH_4 + OH \longrightarrow CH_3 + H_2O)$
 $2\,(CH_3O + O_2 \longrightarrow H_2CO + HO_2)$
 $HO_2 + HO_2 \longrightarrow H_2O_2 + O_2$
 $2\,(CH_3 + O_2 + M \longrightarrow CH_3O_2 + M)$
 $2\,(CH_3O_2 + O \longrightarrow CH_3O + O_2)$
 $2\,(H_2CO + hv \longrightarrow CO + H_2)$
 $2\,(O_3 + hv \longrightarrow O + O_2)$
 $H_2O_2 + hv \longrightarrow OH + OH$
 Net: $2\,CH_4 + 2\,O_3 \longrightarrow 2\,H_2 + 2\,H_2O + O_2 + 2\,CO$ | 13.697 | 14.51 | 5.5 |
| $D_{2.3}$ | $HO_2 + HO_2 \longrightarrow H_2O_2 + O_2$
 $2\,(OH + O_3 \longrightarrow HO_2 + O_2)$
 $H_2O_2 + hv \longrightarrow OH + OH$
 Net: $2\,O_3 \longrightarrow 3\,O_2$ | 22.404 | 22.442 | 7.5 |
| $D_{2.4}$ | $O_3 \longrightarrow O_{3trpt}$
 Net: $O_3 \longrightarrow O_{3trpt}$ | 83.208 | 83.195 | 10.5 |
| $D_{2.5}$ | $O_3 + hv \longrightarrow O + O_2$
 $O \longrightarrow O_{trpt}$
 Net: $O_3 \longrightarrow O_2 + O_{trpt}$ | 33.453 | 33.333 | 15.5 |
| $D_{2.6}$ | $O(^1D) + H_2O \longrightarrow OH + OH$
 $OH + HO_2 \longrightarrow H_2O + O_2$
 $OH + O_3 \longrightarrow HO_2 + O_2$
 $O_3 + hv \longrightarrow O(^1D) + O_2$
 Net: $2\,O_3 \longrightarrow 3\,O_2$ | 27.694 | 28.375 | 16.5 |
| $D_{2.7}$ | $NO + O_3 \longrightarrow NO_2 + O_2$
 $NO_2 + O \longrightarrow NO + O_2$
 $O_3 + hv \longrightarrow O + O_2$
 Net: $2\,O_3 \longrightarrow 3\,O_2$ | 49.204 | 49.202 | 28.5 |

| | | | | |
|---|---|---|---|---|
| $D_{2.8}$ | $\text{HO}_2 + \text{O} \longrightarrow \text{OH} + \text{O}_2$
$\text{OH} + \text{O}_3 \longrightarrow \text{HO}_2 + \text{O}_2$
$\text{O}_3 + \text{hv} \longrightarrow \text{O} + \text{O}_2$
$\text{Net: } 2\,\text{O}_3 \longrightarrow 3\,\text{O}_2$ | 12.92 | 12.891 | 23.5 |
| $D_{2.9}$ | $\text{O}(^1\text{D}) + \text{N}_2 \longrightarrow \text{O} + \text{N}_2$
$\text{NO} + \text{O}_3 \longrightarrow \text{NO}_2 + \text{O}_2$
$\text{NO}_2 + \text{O} \longrightarrow \text{NO} + \text{O}_2$
$\text{O}_3 + \text{hv} \longrightarrow \text{O}(^1\text{D}) + \text{O}_2$
$\text{Net: } 2\,\text{O}_3 \longrightarrow 3\,\text{O}_2$ | 39.97 | 39.973 | 38.5 |
| $D_{2.10}$ | $\text{O}(^1\text{D}) + \text{N}_2 \longrightarrow \text{O} + \text{N}_2$
$\text{HO}_2 + \text{O} \longrightarrow \text{OH} + \text{O}_2$
$\text{OH} + \text{O}_3 \longrightarrow \text{HO}_2 + \text{O}_2$
$\text{O}_3 + \text{hv} \longrightarrow \text{O}(^1\text{D}) + \text{O}_2$
$\text{Net: } 2\,\text{O}_3 \longrightarrow 3\,\text{O}_2$ | 10.943 | 10.946 | 42.5 |

Table 1: Comparison of $O_3$ destruction pathways for two different methods of solving equation 17 in the main manuscript: Using the first solution found by Scipy's lsq_linear and using the procedure described in Lehmann(2004). We only show the first 10 destruction pathways shown in figure 1.

3. Table 3: Pathways D2,1, P2,3, P2,4, P2,5: In the troposphere CH3O2 has a short chemical lifetime (usually $< 1$ min). Therefore we would expect that its fate is controlled by local chemistry, not by transport. Nevertheless, the pathways mentioned above involve transport of CH3O2. Is it possible that this transport of CH3O2 is a numerical artifact, resulting from its calculation according to Eq. (46)? There the (possibly small) contribution of transport Ti is calculated as the difference of (possibly larger) chemical terms. Pathway D2,5: The same type of question applies to D2,5: This pathway involves transport of atomic oxygen, which has a short chemical lifetime in the altitude region indicated (around 15.5 km).

**Response**: In the model we used $\text{CH}_3\text{O}_2$ has a tropospheric number density that ranges between $10^6$ and $10^{11}$ molecules/$cm^3$, and a lifetime against chemical destruction that ranges between 1 and 170 days (figure 2, first row). That is why $\text{CH}_3\text{O}_2$ transport rates are important in our results. Figure 2 (first row) presents the different terms of equation 46 for $\text{CH}_3\text{O}_2$ between 0 and 20km. The chemical production and destruction do not balance the $\text{CH}_3\text{O}_2$ rate of concentration change, so we assume that transport contributes to the remaining molecules to achieve balance. As a consequence, we do not think the presence of $\text{CH}_3\text{O}_2$ transport in some pathways is a numerical artifact. This might be the result of an incomplete representation of the $\text{CH}_3\text{O}_2$ chemistry in the *photochem* model. We did not find references with observational $\text{CH}_3\text{O}_2$ concentrations to compare to the model we are using. Are you aware of any?

We show the different terms of equation 46 for O between 0 and 35km in figure 2 (second row). The chemical production and destruction do not balance the O rate of concentration change between 12 and 35 km. As a consequence, O transport shows as important in these altitudes in our results using the *photochem* model.

[Figure]

Figure 1: Comparison of O₃ destruction pathways for two different methods of solving equation 17 in the main manuscript: Using the first solution found by Scipy's lsq_linear (left column) and using the procedure described in Lehmann(2004) (right column).

[Figure]

Figure 2: Number density (first column), lifetime against chemical destruction (second column) and terms of equation 46 in the main manuscript (third column) for $CH_3O_2$ (first row) and O (second row). The label definitions in the third row are $\frac{d\rho_i}{dt}$ (rate of concentration change), $\Pi_i - L_i$ (chemical production minus destruction), $T_i$ (transport rates). We do not show rainout rates because they are zero or very close to zero.

4. How many of the 1281 reactions have a rate $> f_{min}$, so that they actually take part in the formation of pathways?

   **Response**: The number of reactions with rate $> f_{min}$ varies with altitude, and ranges from 78 to 132. We will update the text to mention this.

5. $CH_4$ is not photochemically produced in Earths atmosphere (G. Brasseur and S. Solomon: Aeronomy of the Middle Atmosphere, Springer, Dordrecht, 2005: p. 296). If this is true also in your model, then $f_{min}$ as defined in lines 388-389 will be zero.

   **Response**: Our reaction system also includes a pseudo-reaction for $CH_4$ supply from transport. For this reason, $f_{min}$ is different from zero. We will clarify this in the manuscript.

6. How long is one time step? **Response**: The *photochem* model uses a solver that uses a variable timestep (CVODE BDF method created by Sundials Computing). In our simulation the timestep varies from $10^{-5}$s to $10^{12}$s. In our analysis we only output the model results when the simulation time is greater than $10^{11}s$. We will clarify this in the manuscript.

---

## Author Response (AR1)

**Final author response**

**Daniel Garduno Ruiz, Colin Goldblatt, Anne-Sofie Ahm**

December 2, 2024

Dear Editor:

Here we provide responses to all of the comments made by the reviewer of the manuscript and specify the changes made to the manuscript.

In summary, there are two main changes to the manuscript. First, we updated the simple example we used to explain how the algorithm works. A person pointed us to a wrong reaction in our simple example (H2O + hv  $\longrightarrow$  OH + O which should be H2O + hv  $\longrightarrow$  OH + H). We updated the simple example with new reactions to correct this error and to make it simpler. Second, we included a discussion sub-section in section 4 of the manuscript to discuss the anomalous CH3O2 pathways in our results and the validation of Chempath.

**Responses to comments**

• Before the actual determination of pathways, the original algorithm checks the balance of the input data (= output of a chemical model) (Lehmann, 2002, Section 3.1): Are the reaction rates consistent with the concentration changes calculated by the model? This is an essential step to detect (and possibly correct) imbalances, which may arise from numerical inaccuracies (e.g., because of a large time step in the chemical model). Balanced input data are indispensable for the actual pathway formation. The authors do not mention how they solve this problem in their implementation.

**Response**: As mentioned in the open discussion, we assume that the reader ensures that mass balance is achieved in their model output. In response to this comment, we updated the code to include a function that displays a warning if any of the species are not balanced by the reactions. We added the following text to section 3 of the manuscript to describe this new function:

"Before the construction of pathways it is essential to ensure the concentration changes of all species are balanced by the reaction's production and destruction (equation 1). Our implementation does not try to correct for imbalances, but we include a function that displays a warning if mass balance is not fulfilled. The warning is displayed if the unbalance is greater than a relative tolerance of  $1 \times 10^{-3}$  if the concentration change is greater or equal to 1 molecule/ $cm^3$ , and to an absolute tolerance of  $1 \times 10^{-3}$  if the concentration change is lower than 1 molecule/ $cm^3$ . We use an absolute tolerance for concentration changes lower than 1 molecule/ $cm^3$  because we consider that concentration changes lower than  $1 \times 10^{-3}$ molecules/ $cm^3$  are unimportant. If this warning is displayed, the model output should be checked for potential problems or corrected (see (Lehmann, 2002) for an example of how to do this)." • How does the algorithm choose a splitting of a pathway into sub-pathways if this splitting is not unique? Does the formulation we choose the solution that minimizes the most equation 19" mean that a solution of equation 19 is chosen? Although it is true that the optimization problem (19) has a global minimum (line 221), it is not guaranteed that there is a unique solution. In fact, if Equation (17) has multiple solutions, this will also be the case for the related optimization problem (19).

**Response**: As mentioned in the open discussion, we chose the first solution that minimizes equation 19 found by Scipy's lsq\_linear algorithm. As a response to this comment, we updated the code to include the option to solve equation 17 of the manuscript with the procedure described in section 5.5.1 of Lehmann (2004). We find similar results with both methods. We updated line 217 of the manuscript to describe this new potion:

"Our implementation includes two options to solve equation 16. The first option uses Scipy's "lsq\_linear" function, minimizing the expression:

$$0.5||Ax - b||^2 \text{ with constraints } 0 \le x < \infty, \tag{1}$$

where ||x|| is the norm of x,  $A = [[x'_{je}]]$ ,  $x = [w_e]$  and  $b = [x_{je}]$ . When there are multiple solutions to equation 16, we choose the first solution that minimizes equation 18 found by the "lsq\_linear" algorithm. The second option to solve equation 16 implements the method proposed in section 5.2.2 of (Lehmann, 2004). This method chooses the solution that produces more probable pathways in the sense that this solution produces simpler pathways with higher rates compared to other solutions. Both methods of solving equation 16 produce similar results."

• Table 3: Pathways D2,1, P2,3, P2,4, P2,5: In the troposphere CH3O2 has a short chemical lifetime (usually < 1 min). Therefore we would expect that its fate is controlled by local chemistry, not by transport. Nevertheless, the pathways mentioned above involve transport of CH3O2. Is it possible that this transport of CH3O2 is a numerical artifact, resulting from its calculation according to Eq. (46)? There the (possibly small) contribution of transport Ti is calculated as the difference of (possibly larger) chemical terms. Pathway D2,5: The same type of question applies to D2,5: This pathway involves transport of atomic oxygen, which has a short chemical lifetime in the altitude region indicated (around 15.5 km).

**Response**: As mentioned in the open discussion,  $CH_3O_2$  and O are present in these pathways because of an incomplete representation of their chemistry in the reaction system we used. We updated the manuscript to include a new section to discuss the presence of these species in these pathways.

• 1 As mentioned above, the authors describe a re-implementation of the (existing) pathway analysis program of Lehmann (2004) in Python. This is said correctly in the body of the text, but it would be good to state this more clearly in the abstract (and, maybe, conclusions) for the hasty reader.

**Response**: We modified the abstract to make this clear, including the following sentence "*Chempath* is a Python re-implementation of the algorithm developed by Lehmann (2004)."

• 36-37 This leads to a difficult discussion. Available on request may also be considered as a form of open - with the additional advantage that through the personal contact the user can obtain all support needed. Anonymous download of a program is risky if the user does not understand perfectly the functionality and limitations of the program, which may be hard to achieve even if there is a good documentation. I understand that the authors want to provide a reason for their re-programming effort, but I think that the formulation in line 38 is sufficient for that.

**Response**: We deleted these lines.

• 64-72 Strictly speaking, ppb is the unit for mixing ratio, not concentration.

Table 1: Row sij : For readers familiar with chemical systems you might add stoichiometric matrix.

Table 1: Rows  $\hat{r}_j$ ,  $\hat{p}_i$ ,  $\hat{d}_i$ : The reader might be surprised by these definitions, because deleted pathways have not been mentioned before. An earlier mentioning of deleted pathways would also be beneficial for Section 2.5 (describing the calculation of rates of deleted pathways), which is placed before Section 2.7, where deleted pathways are introduced.

**Response**: We changed concentration to mixing ratio. We added stoichiometric matrix to table 1. We also added the following paragraph to mention deleted pathways before table 1:

"In large reaction systems, it might not be possible to construct all the pathways of the system because the number of pathways could be computationally unmanageable. The algorithm includes the option to delete unimportant pathways to avoid the construction of an unmanageable number of pathways and enhance the computation time. However, the algorithm includes variables to keep track of the rates of these deleted pathways."

• 81 (and several other lines) The official symbol four hour is h. Please insert blanks between numbers and units

**Response**: We changed hr to h and inserted blanks between numbers and units.

81 The notation ∑j=15[1,0,1,0,0] · [1,05,1.5,5,0.1] does not make sense, since there is no j in the terms after the sum. A workaround might consist in writing the sum explicitly (1 ů 1 + 0 ů 0.5 + ...) or as scalar product.

**Response**: We updated the text to include the recommended notation.

• 83 I would not mention the reaction system together with in two consecutive time steps (it the same for both time steps). 84 mean reaction rates: in two consecutive time steps or between two consecutive time steps or within one time step?

**Response**: We changed the sentence to:

"The algorithm requires four inputs from a chemical kinetics model: the species concentrations and the model time in two consecutive model times t and t + dt, the mean reaction rates in the time step dt, and the reaction system with  $n_r$  reactions between  $n_s$  species."

• 84 It may happen that the concentration of a species at two points in time t1 and t1 + t is not sufficient as input, but information on the concentration between t1 and t1 + t, e.g. its mean, is also needed: For instance, if you analyse tropospheric chemistry over a full day

(from midnight t1 to the next midnight t1 + t, t = 24 h), you will obtain [OH] zero at t1 and t1 + t (although OH is present during the day), which leads to a wrong estimate of the lifetime of OH in [t1, t1 + t] (needed in line 94).

**Response**: We included the following sentence to clarify that the two points in time must be chosen in a way that the processes one is interested in understanding are resolved by the time step:

"The consecutive model time steps must be the time steps in which the solver obtains a solution for the system of equations. The algorithm could also be applied in two model times that are not consecutive, but the time step must be small enough to resolve the processes one is interested in understanding."

• 90  $f_j = r_j$  instead of  $f_{kinit} = r_j$ ?

**Response**: We included the suggested change.

• Fig. 1: Box 8: Recomputation of variables: Which variables? How?

**Response**: This is explained in section 2.9

• 117 ... produced (or consumed) by one pathway is consumed (or produced) by another pathway might be a bit clearer than the present formulation.

**Response**: We included the suggested change.

• 123-124 As these operations are carried out for each pathway separately (i.e. g may differ from pathway to pathway), the sentence should be formulated in singular: The multiplicities xij of a new pathway ...The rate of the new pathway is multiplied...

**Response**: We included the suggested change

• 125, 130 (and several other lines) Throughout the manuscript the authors use identical denotations for elements of a matrix (or vector) and the whole matrix (or vector). Although the reader may guess what is meant, I recommend a stricter notation, especially since the number of indices does not always indicate the dimension of the object, e.g. xjn in line 130 denotes a vector.

**Response**: We updated our notation, using two square brackets around variables that denote matrices (for example  $[[s_{ij}]]$ ) and one around variables that denote vectors (for example  $[r_j]$ ).

• 134, 197, 256, 287: Why similar to instead of has the following form?

**Response**: We replaced similar with has the following form.

• 143 in the previous step from the previous step (or omit completely)?

**Response**: We omitted these words.

• 153 Eq. (10):  $\sum_k \rightarrow \sum_e$  (3 times)

**Response**: We included the correction

• 179-181 Lines 176-181 describe the deletion of pathways that have been used (i.e. connected to other pathways). This step does not involve any consideration of deletion due to rates < fmin. Why are  $\hat{r_i}$ ,  $\hat{p_i}$ ,  $\hat{d_i}$  mentioned nevertheless? (And what exactly does this case refer to?)

**Response**: We omitted the sentence

- 218 minimizing the equation → minimizing the expression
  Response: We included the suggested change
- 219  $x \le \infty \to x < \infty$

**Response**: We made the correction

• 220 A comparison with the variables in (17) indicates that:  $A = ((x'je))j = 1, ..., n_r, e = 1, ..., n_e \ b = (xjc)j = 1, ..., n$

**Response**: Thanks for noticing this error. We included the correction.

• 277  $0.007 \rightarrow 0.0073$  (in order to be consistent with Eq. (30))

**Response**: We used the same number of decimals in the corrected simple example.

• 304 this contribution  $\rightarrow$  the contribution

**Response**: We included the correction

• 306 Eq. (42):  $Sb \rightarrow Si$  (2 times)

**Response**: Thanks for noticing this error. We included the correction.

• 308-309 It seems that For example, ... (43) should be placed directly after Eq. (42), not after text about deleted pathways.

**Response**: We moved the text about deleted pathways to the end of the section.

• 309 Eq. (43): This equation involves element-wise multiplication of two vectors (not scalar product). Perhaps this should be said explicitly.

**Response**: We included the following text after equation 42 to clarify this: Expression 42 involves the element-wise multiplication of two vectors.

• Table 2: HV  $\rightarrow h\nu$  (several times)

**Response**: We included the correction

353-354 setting it as a fraction of the rate of production of the species...: Here production refers to the total production by all reactions? If so, you might emphasize this, in order to avoid confusion with the production by pathways in Table 3. In general, this way of choosing fmin may still require further trial and error: If the species of interest (Si) is involved in zero cycles with large rates (e.g. O3 O + O2), then the rate of the (total) production of Si will be much larger than the net production or destruction, which shall be explained by pathways. This may have the consequence that an originally chosen fraction (i.e. fmin) may turn out to be too large and must be reduced.

354 Shouldnt destruction rates also be taken into account, e.g. for species like CH4 that is only destroyed in Earths atmosphere?

**Response:** We modified the text to clarify that we refer to the total production by all reactions, and that setting  $f_{\min}$  as a fraction of this production might still require further

trial and error. We agree that destruction rates should be taken into account for species like CH4:

"Second, the user needs to choose a minimum rate of pathways  $f_{\min}$ . This can be done by trial and error, or setting it as a fraction of the rate of total production or destruction by the reactions of the species the user is interested in finding pathways for. However, this way of setting  $f_{\min}$  might still require further trial and error to find an appropriate fraction of the total production or destruction by all reactions."

• 360 How many of the 1281 reactions have a rate > fmin, so that they actually take part in the formation of pathways?

**Response**: The number of reactions with rate >  $f_{min}$  varies with altitude, and ranges from 78 to 132. We included the following text:

"The number of reactions with rate >  $f_{min}$  varies with altitude, and ranges from 78 to 132."

• 364 Please explain to the reader the idea behind the reduction of the O2 surface flux. Which processes in the model lead to a removal of O2 (eventually balancing the source by the surface flux)? What is the time scale of these processes?

**Response**: We included the following text to explain this:

"The idea behind the reduction of the  $O_2$  surface flux is to create a perturbation that causes concentration changes to explore with *Chempath*. The concentration of  $O_2$  in the model is controlled mainly by the surface flux and by oxidation of reduced species like CH4, CO, and H2 in a timescale of millions of years (the estimated lifetime for  $O_2$  in the modern atmosphere is ~2 million years (?))."

- Response:
- 365 every time step: How long is one time step?

**Response**: We included the following text to answer this question:

"The *photochem* model uses a solver with an adaptive timestep (CVODE BDF method created by Sundials Computing). In our simulation the timestep varies from  $10^{-5}$ s to  $10^{12}$ s. We only output the model results when the simulation time is greater than  $10^{11}s$ ."

368-370 ... if we want to know what are the chemical mechanisms that contribute to this O3 loss, we need to use the pathway analysis program. We apply Chempath to the photochem model output to gain insight into the chemical reaction chains that destroy O3 in this model run.: This formulation sounds as if the O3 destruction pathways directly explain the O3 decrease (probably of a few ppb / million years ~ 106 ppb/y) occurring in the model run after the reduction of the O2 surface flux. However, this is not the case. As Ox (= O3 + O + O(1D)) has a chemical lifetime of 1 year below 100 km (and much less in the middle atmosphere) (e.g., G. Brasseur and S. Solomon: Aeronomy of the Middle Atmosphere, Springer, Dordrecht, 2005: Fig. 5.3), it will be close to equilibrium in your million-year long model run, i.e. the concentration is determined by the production rate (strongly dependent on the changing [O2]) and the time scale of destruction. By the way, these arguments may serve as motivation for showing production and destruction pathways later on (Table 3).

**Response**: Thanks for sharing these arguments. We agree with your analysis. We modified this paragraph to clarify that both ozone production and destruction pathways are important to understand the O3 concentration change:

"We apply *Chempath* to the *photochem* model output to gain insight into the chemical reaction chains that produce and destroy  $O_3$  in this model run."

• 372-373 vertical transport production and destruction e.g., supply and removal by vertical transport?

**Response**: Thanks for noticing this error. We included the correction.

• Figure 2 (upper left panel): m means milli =  $10^3$

**Response**: We changed my to million years.

• 374 Eq. (45):  $\text{Li} \Rightarrow \text{Di}$

**Response**: Thanks for noticing this error. We included the correction.

• 376 Does production by rainout mean that evaporation of rain and release of trace gases from the liquid phase to the gas phase is included in the model? According to Eq. (47) this seems not to be the case.

**Response**: Thanks for noticing this error. We updated the text to only say destruction by rainout.

• 381 You might include di/dt in the list of values obtained from the model.

**Response**: We already mentioned that we obtained the number density and the model time at two consecutive model times to calculate this expression.

• 384 Eq. (47): It seems that supplies and removes should be interchanged.

**Response**: Thanks for noticing this error. We included the correction.

• 388-389 CH4 is not photochemically produced in Earths atmosphere (G. Brasseur and S. Solomon: Aeronomy of the Middle Atmosphere, Springer, Dordrecht, 2005: p. 296). If this is true also in your model, then fmin as defined in lines 388-389 will be zero. Please clarify.

**Response**: Our reaction system also includes a pseudo-reaction for  $CH_4$  supply from transport. For this reason,  $f_{min}$  is different from zero. We modified this sentence to make this clear:

We prescribe a variable minimum pathway rate  $f_{\min}$  that we calculate as the minimum of the chemical production by reactions (including transport pseudo-reactions) of O2, O3, CO, H2 and CH4 divided by 1000

• Figure 3: It would be nice to use similar colours (or additional symbols) to indicate pathways of the same family (HOx, NOx etc.).

**Response**: We included additional symbols in the legend of the figure to group the pathways into five categories: Oxidation, Chapman-like, photolysis, HOx, and NOx pathways. We also updated the figure caption to explain the additional symbols.

• Figure 3, Table 3: It might be more logical to present production pathways before destruction pathways.

**Response**: We changed the order of the pathways

• Figure 3, Table 3 and related text: All pathways have the same first index 2. Therefore it might be omitted in the manuscript (probably it results from the fact that O3 is species no. 2 in the model).

**Response**: We omitted the index

• Table 3: O1D *Rightarrow* O(1D) (several times)

**Response**: We made the correction

• Table 3: It seems that the algorithm does not attempt to order the reactions in a way that the flux of molecules can be easily followed by the user. I suggest to do that by hand" (and clearly state that it was done in this way); e.g. pathway D(2)2 might be reordered in the following way:

**Response**: We ordered the reactions by hand and included the following sentence in the caption of table 3: Our algorithm does not yet have the functionality to automatically order the reactions to easily follow the flow of molecules. We ordered the reactions in all the pathways by hand.

• Figure 4: Upper panel: Delete s" in the unit of the number density

**Response**: We included the correction

- Figure 4: Caption: middle panel and bottom panel does not coincide with the figure above. **Response**: We updated the figure to coincide with the caption description.
- Figure 4: Caption: tables  $3 \Rightarrow$  table 3"

**Response**: We included the correction

**References**

- Lehmann, R.: Determination of Dominant Pathways in Chemical Reaction Systems: An Algorithm and Its Application to Stratospheric Chemistry, Journal of Atmospheric Chemistry, 41, 297–314, https://doi.org/10.1023/a:1014927730854, 2002.
- Lehmann, R.: An Algorithm for the Determination of All Significant Pathways in Chemical Reaction Systems, Journal of Atmospheric Chemistry, 47, 45–78, https://doi.org/10.1023/b: joch.0000012284.28801.b1, 2004.

---

## Referee Report (RR1)

**Review of**

**"Chempath 1.0: An open-source pathway analysis program for photochemical models" (revised version of 3 December 2024) by D. Garduño Ruiz et al.**

(Numbers refer to line numbers in the manuscript version with tracked changes.)

**Major comments**

- 367 An essential part of the original algorithm, the balancing of concentration changes and reaction rates, is missing in the implementation presented in the manuscript. The assumption that "the reader ensures that mass balance is achieved in their model output" (final author response and line 372) is unrealistic.
- 438 In their model, the authors enforce the balance mentioned above by an unphysical "trick": They calculate the transport term as the difference of the concentration change and the reaction rates (including wet removal) (Eq. 47). As a consequence, the balance between concentration change, reaction rates (including wet removal) and transport is automatically guaranteed. However, this procedure implies that all numerical errors in the calculation of concentration changes and reaction rates are interpreted as transport effects.

It is likely that this procedure leads to artificial pathways involving transport, e.g. D5. The reason for D5 given by the authors in the final response (incomplete representation of the chemistry) is less probable: The model contains the main O loss reaction  $O + O_2 + M \rightarrow O_3 + M$ . This ensures a short chemical lifetime and a small mixing ratio of atomic oxygen, even if additional reactions are missing. As a consequence, transport fluxes of O should be small (compared to the O3 loss rate to be explained).

422-425 As  $O_x$  (=  $O_3 + O + O(^1D)$ ) has a chemical lifetime of  $\leq 1$  year below 100 km (and much less in the middle atmosphere), it will be close to equilibrium in the million-year long model run, i.e. the concentration is determined by the production rate (dependent on the changing  $[O_2]$ ) and the time scale of destruction. By determining production and destruction pathways of  $O_3$  the authors show how this equilibrium is maintained at selected points in time. This is a valid analysis. However, it is not the answer to the problem that the authors announced to solve: "... know what are the chemical mechanisms that explain the  $O_3$  concentration change ... as a consequence of the decrease in the  $O_2$ surface input flux". In addition to the argument just mentioned, it can be noticed that the model has reached an equilibrium state and has thus "forgotten" the initial concentrations, corresponding to the unperturbed  $O_2$  surface flux, at the times of the analysis (4.1 million years in Table 3 and 4.5 million years in Fig. 3) (cf. Fig. 4).

**Details**

- 72 "concentration"  $\Rightarrow$  "mixing ratio"
- 81, 204 "enhance"  $\Rightarrow$  "reduce"?
  - 90 "and the model time in"  $\Rightarrow$  "at" (to avoid mentioning "model time" twice)?
  - 93-94 These two sentences sound contradictory ("must be ..."  $\leftrightarrow$  "could also be ..."). The length of the time interval of the pathway analysis can be chosen idependently from the time step of the solver.
    - 206 Equation (14): Does the multiplication mean scalar product? The notations " $[m_{iq}] > 0$ " and " $[m_{iq}] < 0$ " are problematic: a vector is compared to a number.
    - 230 Equation (16): What kind of product (vector times matrix)?
    - 244 "section 5.2.2"  $\Rightarrow$  "section 5.5.2"
    - 261 " $m_{ik}$ "  $\Rightarrow$  "[[ $m_{ik}$ ]]"
    - 315 Equation (33): " $[m_{i1}] > 0$ " and " $[m_{i1}] < 0$ ": vector compared to number (cf. 206). Moreover, these expressions should not be placed within an equation.
    - 347 The explanation "The expression 42 involves ... (44)" should appear directly after Equation (42).
    - 444 "updated"  $\Rightarrow$  "augmented"?
  - Fig. 3 The abbreviation "Ox" ( $\doteq$  "Oxidation" of what?), standing near "NOx" and "HOx", might be easily confused with "Ox" ( $\doteq$  O3 + O + O(1D)).
    - 500 "shows"  $\Rightarrow$  "show"

---

## Referee Report (RR2)

**Review of**

**"Chempath 1.0: An open-source pathway analysis program for photochemical models" (revised version of 7 April 2025) by D. Garduño Ruiz et al.**

(Numbers refer to line numbers in the manuscript version with tracked changes.)

**Major comment**

The authors introduce a way of tracking the consequences of imbalances in the input data. However, they do **not** provide a remedy for such imbalances.

- 379 As the authors only provide a tracking of imbalances, the headline of Section 3.2 "Balancing of concentration changes" is misleading and should therefore be changed.
- 382-383 "... we assume that the difference between concentration changes and the total production by all reactions is due to the solvers numerical error...": In most applications the concentration change is calculated by the solver (with high accuracy), whereas the reaction rates are calculated separately after the main model run (from data with less temporal resolution than the internal calculations by solver), which introduces larger errors (and justifies a subsequent modification of the rates).
  - 395 As stated correctly by the authors, they request that the user ensures the balance of the input data. As explained above, this may be problematic for some users. Therefore a corresponding information ("warning") should be included in the abstract and introduction together with a notice that a part of the original algorithm of Lehmann (2004) was not implemented.

**Details**

- 76-371 Please check carefully throughout the whole manuscript whether the square brackets really denote matrices or vectors as defined in line 76 or just single elements of a matrix or vector (contrary to their definition in line 76), for instance:
  - Table 1:  $[dc_i]$  should be explained as "**vector** of the concentration changes..." (or the square brackets would have to be removed).
  - Table 1 (last line): If  $[\tilde{c}_i]$  and  $[d_i]$  denote vectors, then  $[\tau_i]$  must be calculated by element-wise division.

- Eq. (1): This looks like a product of a matrix and a vector, but then the summation would not be correct (and there is one closing bracket too much).
- 200: "where  $i = 1, ..., n_s$ " seems to denote  $n_s$  individual equations, which contradicts the vector notation in Eq. (14).
- 218: You mention **one** vector  $[w_e]$ , but calculate the sum over  $n_e$  such vectors in Eq. (16). (Is the transpose symbol in Eq. (16) correct?)
- 77-79 "keeping only the positive values"  $\Rightarrow$  "keeping only the positive values and zero"? (analogously for negative values)
  - 366 I suggest to keep the deleted "as branching points" for clarity.
- 412-413 "constant rates" or "constant rate constants"? "these rates" or "these rate constants"?
  - 426 "We run Chempath ... in all the model times...": This sounds like points in time. What is the length of the time intervals analysed, corresponding to dt in Table 1?
  - 472 "We run Chempath ... at 32 time points...": cf. previous question.
  - 473 Does this sentence mean that the model run was 1 million years long (cf. also line 439)? If so, why do Table 4 and Figs. 3, 4, and 5 show results for "time = 1.16 million years"?
  - 505 Probably the sequence of causes and effects is: decrease of the  $O_2$  input flux  $\Rightarrow$  decrease of the  $O_3$  production rate  $\Rightarrow$  decrease of the  $O_3$  concentration  $\Rightarrow$  decrease of the  $O_3$  loss rate
- 520-521 Under conditions of present Earth (troposphere), the reaction  $CH_3O_2$ + NO  $\rightarrow$  ..., which is included in your model, would ensure a short lifetime of  $CH_3O_2$ . However, the mixing ratio of NO in your model seems to be extremely small: From the lowest points in Figure 3 (red line) we can obtain the following rough estimate of the NO mixing ratio  $r_{NO}$ :

$$r_{\rm NO} = [{\rm NO}]/[{\rm M}]$$

 $\leq [{\rm NO}]/(5 \cdot [{\rm O}_2])$
 $\approx 4 \cdot 10^4/(5 \cdot 2 \cdot 10^{18})$
 $= 4 \cdot 10^{-15}$
 $= 4 \cdot 10^{-6} \text{ ppb}$

521 "these species": which species (or why plural)?

Table 4 The unit of the rates  $(\text{molec}/(\text{cm}^2 \cdot \text{s}))$  seems to imply vertically integrated rates. However, the figure caption says "... rates ... correspond to the height at which the pathways contribute the most..."

**Typos**

- 219 "weighs"  $\Rightarrow$  "weights"
- 425 "figure 2e"  $\Rightarrow$  "figure 2f"
- 503 " $P_{2.9}$  to  $P_{2.18}$ "  $\Rightarrow$  " $D_{2.9}$  to  $D_{2.18}$ "

---

## Author Response (AR2)

**Author response**

**Daniel Garduno Ruiz, Colin Goldblatt, Anne-Sofie Ahm**

April 2, 2025

Dear Dr. Sander:

Here we provide responses to all of the comments made by the reviewer and specify the changes made to the manuscript. We have addressed all the comments and revised the manuscript accordingly.

**Responses to reviewer comments**

• An essential part of the original algorithm, the balancing of concentration changes and reaction rates, is missing in the implementation presented in the manuscript. The assumption that "the reader ensures that mass balance is achieved in their model output" (final author response and line 372) is unrealistic.

**Response**:**

We updated the way we do our pathway analysis to balance the concentration changes and reaction rates including error pseudo-reactions that produce or destroy a species at the rate required to achieve balance. We included the following text in the manuscript to describe this way of balancing the concentration changes:

"Before the construction of pathways, it is essential to ensure that the concentration changes of all species are balanced by the reaction's production and destruction (equation 1). The balance might not be fulfilled due to numerical errors. To quantify this problem, we assume that the difference between concentration changes and the total production by all reactions is due to the solver's numerical error, and we include error pseudo-reactions in the chemical system that produce or destroy a species at the rate required to fulfill the balance. We include the error pseudo-reactions in the construction of pathways. After the pathway construction finishes, we delete the pathways containing error pseudo-reactions, updating  $\tilde{r_j}$ ,  $\tilde{p_i}$  and  $\tilde{d_i}$ . We also include variables similar to  $\tilde{r_j}$ ,  $\tilde{p_i}$  and  $\tilde{d_i}$  to track the rates of the pathways containing error pseudo-reactions. This approach gives the user information on how important the numerical error is in explaining the concentration changes. Ideally, the pathways containing error pseudo-reactions will not contribute significantly to the concentration change one is interested in understanding."

• In their model, the authors enforce the balance mentioned above by an "unphysical trick": They calculate the transport term as the difference of the concentration change and the reaction rates (including wet removal) (Eq. 47). As a consequence, the balance between concentration change, reaction rates (including wet removal) and transport is automatically guaranteed. However, this procedure implies that all numerical errors in the calculation of concentration changes and reaction rates are interpreted as transport effects. It is likely that this procedure leads to artificial pathways involving transport, e.g. D5. The reason for D5 given by the authors in the final response (incomplete representation of the chemistry) is less probable: The model contains the main O loss reaction  $O + O_2 + M \longrightarrow O_3 + M$ . This ensures a short chemical lifetime and a small mixing ratio of atomic oxygen, even if additional reactions are missing. As a consequence, transport fluxes of O should be small (compared to the  $O_3$  loss rate to be explained).

**Response:**

We updated the way we do our pathway analysis in the *photochem* model to retrieve the transport rates directly from the model instead of using an inversion to calculate transport with equation 47. Figure 2 shows a comparison of transport rates calculated with equation 47 and transport rates directly retrieved from the model for different species. For some species, the inversion works very well, for example ( $CH_3O_2, CH_4, O_3$ ), but for other species, the inversion does not perform well (for example O). This is likely due to the numerical error, as the reviewer argues.

Using the transport rates retrieved from the model results in an imbalance between the concentration changes and the reaction rates. To balance the concentration changes, we assume that the difference between concentration changes and the right-hand side of equation 47 is due to the numerical error, and we include error pseudo-reactions that produce or destroy a species at the rate required to balance the concentration changes. The numerical error is unavoidable, and this way of balancing the concentration changes can give the user an idea of how much the error contributes to explaining the concentration changes.

We revised our analysis with this updated method. Pathway D5 is no longer detected with this new approach. This result suggests that the reviewer was right, and this pathway was a consequence of interpreting numerical errors as transport. In our new analysis, the contribution of the pathways containing error pseudo-reactions to O3 production or destruction was less than 1% (figure 2 shows an example of this). We updated the manuscript to describe the updated method and results.

We also update equation 47 to include a vertically distributed input flux  $(F_i)$  for some species (for example SO2)

As Ox (= O3 + O + O(1D)) has a chemical lifetime of 1 year below 100 km (and much less in the middle atmosphere), it will be close to equilibrium in the million-year-long model run, i.e. the concentration is determined by the production rate (dependent on the changing [O2]) and the time scale of destruction. By determining production and destruction pathways of O3 the authors show how this equilibrium is maintained at selected points in time. This is a valid analysis. However, it is not the answer to the problem that the authors announced to solve: "... know what are the chemical mechanisms that explain the O3 concentration change ... as a consequence of the decrease in the O2 surface input flux".

In addition to the argument just mentioned, it can be noticed that the model has reached an equilibrium state and has thus forgotten the initial concentrations, corresponding to the unperturbed  $O_2$  surface flux, at the times of the analysis (4.1 million years in Table 3 and 4.5 million years in Fig. 3) (cf. Fig. 4).

Figure 1: Comparison of transport rates calculated with equation 47 and transport rates directly retrieved from the model for different species at time=1.16 million years in our *photochem* model run.

---

## Author Response (AR3)

**Author response**

**Daniel Garduno Ruiz, Colin Goldblatt, Anne-Sofie Ahm**

April 27, 2025

Dear Dr. Sander:

Here we provide responses to all of the comments made by the reviewer and specify the changes made to the manuscript. We have addressed all the comments and revised the manuscript accordingly.

**Responses to reviewer comments**

• The authors introduce a way of tracking the consequences of imbalances in the input data. However, they do not provide a remedy for such imbalances.

**Response**: We would like to implement the method of balancing the concentration changes described in Lehman (2022) in a future version of Chempath.

• As the authors only provide a tracking of imbalances, the headline of Section 3.2 "Balancing of concentration changes" is misleading and should therefore be changed.

Response: We changed the headline to "Tracking imbalances due to numerical errors"

• ... we assume that the difference between concentration changes and the total production by all reactions is due to the solver's numerical error...": In most applications the concentration change is calculated by the solver (with high accuracy), whereas the reaction rates are calculated separately after the main model run (from data with less temporal resolution than the internal calculations by solver), which introduces larger errors (and justifes a subsequent modification of the rates).

As stated correctly by the authors, they request that the user ensures the balance of the input data. As explained above, this may be problematic for some users. Therefore a corresponding information ("warning") should be included in the abstract and introduction together with a notice that a part of the original algorithm of Lehmann (2004) was not implemented.

• **Response**: We include the following text in the abstract:

"*Chempath* does not include the balance of concentration changes and reaction rates that Lehmann's algorithm uses to eliminate imbalances due to numerical errors". Instead, *Chempath* quantifies the contribution of these imbalances to the production and destruction of a species.

And the following text in the introduction:

"Our implementation is based on the description of the algorithm in Lehmann (2004). However, there is one difference between Chempath and Lehmann's (2004) algorithm. Our implementation does not include the balance of concentration changes and reaction rates that Lehmann's algorithm uses to eliminate imbalances due to numerical errors. Instead, *Chempath* requires input information about these imbalances to quantify the contribution of numerical errors to the production and destruction of a species."

• Table 1: [dci] should be explained as vector of the concentration changes..." (or the square brackets would have to be removed).

**Response**: We specified that this variable represents a vector in table 1. We did the same with other variables representing vectors.

• Table 1 (last line): If  $[c_i]$  and  $[d_i]$  denote vectors, then  $[\tau_i]$  must be calculated by element-wise division.

**Response:** We specified that  $[\tau_i]$  is calculated by the element-wise division of two vectors

• Eq. (1): This looks like a product of a matrix and a vector, but then the summation would not be correct (and there is one closing bracket too much)

**Response**: We updated this equation to represent the multiplication of a matrix and a vector, to avoid the use of the sum notation. We did the same in all other equations containing sum notations.

• 200: where i = 1...ns seems to denote ns individual equations, which contradicts the vector notation in Eq. (14).

**Response**: We deleted i = 1... ns to be consistent with the vector notation.

• You mention one vector [we], but calculate the sum over ne such vectors in Eq. (16). (Is the transpose symbol in Eq. (16) correct?)

**Response**: We updated this equation to represent the multiplication of a matrix and a vector, to avoid the use of the sum notation. We also include an example of how this equation is fulfilled in our simple example (equation 24 in the corrected manuscript). You can see in that example why we need the transpose symbol.

• "keeping only the positive values" ⇒ keeping only the positive values and zero"? (analogously for negative values).

Response: We deleted the sentences "keeping only the positive/negative values"

• I suggest to keep the deleted "as branching points" for clarity.

**Response**: We re-included this sentence.

• We run Chempath ... in all the model times...": This sounds like points in time. What is the length of the time intervals analysed, corresponding to dt in Table 1?

**Response**: We replaced "model times" with "model time intervals". For all points in time where there is a solution we use Chempath to obtain pathways, using the current point in time and the next point in time at which the solver obtains a solution. We use an adaptive solver with a varying dt. In our simulation dt varies from  $10^{-8}$  seconds to  $10^3$  seconds.

• We run Chempath ... at 32 time points...": cf. previous question.

**Response**: Photochem also uses an adaptive solver with a varying dt. In our simulation dt varies from 0.1 to  $10^{10}$  seconds.

Figure 1: NO mixing ratio in our model run

• Does this sentence mean that the model run was 1 million years long (cf. also line 439)? If so, why do Table 4 and Figs. 3, 4, and 5 show results for .16 million years"?

**Response**: We corrected these sentences to specify that the model was run for 1.2 million years.

• Probably the sequence of causes and effects is: decrease of the O2 input flux ⇒ decrease of the O3 production rate ⇒ decrease of the O3 concentration ⇒ decrease of the O3 loss rate.

**Response**: We modified this sentence to include the sequence suggested here.

• Under conditions of present Earth (troposphere), the reaction CH3O2 + NO ..., which is included in your model, would ensure a short lifetime of CH3O2. However, the mixing ratio of NO in your model seems to be extremely small.

**Response**: Figure 1 shows the NO mixing ratio corresponding to the red line in figure 3 on the manuscript. Its concentration is very small in the troposphere, as you notice. The concentration of NO in the model we are using is determined by chemistry, transport, and deposition. The model does not include lightning or anthropogenic emissions.

- "these species": which species (or why plural)? **Response**: We made the correction.
- The unit of the rates (molec/(cm2/s)) seems to imply vertically integrated rates. However, the figure caption says ... rates ... correspond to the height at which the pathways contribute the most..."

**Response**: We made the correction.